



# A Pan-European, High-Resolution, Daily Total, Fine-Mode and Coarse-Mode Aerosol Optical Depth dataset based on Quantile Machine Learning

Zhao-Yue Chen[1,2], Raul Méndez[1], Hervé Petetin[3], Aleksander Lacima[3], Carlos Pérez García-Pando[3,4] and Joan Ballester[1]

1ISGLOBAL, Barcelona, Spain
2Universitat Pompeu Fabra (UPF), Barcelona, Spain
3Barcelona Supercomputing Center, Barcelona, Spain
4ICREA, Catalan Institution for Research and Advanced Studies, Barcelona,Spain

**Correspondence:** Zhao-Yue Chen (zhaoyue.chen@isglobal.org) and Joan Ballester (joan.ballester@isglobal.org)

**Abstract**

Ambient particulate matter (PM) is a widespread air pollutant, consisting of a mixture of different particle species suspended in the air that negatively affects human health. Given the generally sparse distribution of in-situ PM measurement networks, spatially-resolved PM estimates are typically derived from Aerosol Optical Depth (AOD) obtained from satellites. However, satellite AOD data over land is affected by several limitations (e.g., data gaps; coarser resolution; higher uncertainty; unavailable or unreliable size fraction information), which weakens the relationship between AOD and PM. We have developed a 0.1 degree resolution daily AOD data set over Europe over the period 2003-2020, based on new Quantile Machine Learning (QML) models. The dataset provides reliable full-coverage AOD along with Fine-mode AOD (fAOD) and Coarse-mode AOD (cAOD), based on AERONET (AErosol RObotic NETwork) site observations and climate and air quality reanalyses. Our results show that the three QML AOD products guarantee better quality with an out-of-sample $R^2$ equal to 0.68 for AOD, 0.66 for fAOD and 0.65 for cAOD, which is 23-92%, 11-13% and 115-132% higher than the corresponding satellite or reanalysis products, respectively. Over 88.8%, 80.5% and 88.6% of QML AOD, fAOD and cAOD predictions fall within ±20% Expected Error (EE) envelopes, respectively. Previous studies reported that Europe is one of the regions with the poorest satellite AOD-PM correlation (Pearson correlation coefficient (PCC) around 0.1). Our results show that the three QML products are more correlated with ground-level PMs, especially when they are paired with their corresponding PMs in terms of size: AOD with PM10, fAOD with PM2.5 and cAOD with PM coarse (R=0.41, 0.45 and 0.26, respectively). Our results show that different PM size fractions may be better predicted using different AOD size fractions, instead of total AOD. QML long-term aerosol dataset (and associated models) not only fix some problems of existing AOD data, but also provide better tools to monitor and analyse fine-mode and coarse-mode aerosols in spatial and temporal dimensions, and to further investigate their impacts on human health, climate, visibility, and biogeochemical cycling. The QML datasets can be downloaded from https://doi.org/10.5281/zenodo.7756570 (Chen et al., 2023).





## 1. Introduction

According to the latest report of the Global Burden of Disease (Institute for Health Metrics, 2020), ambient particulate matter (PM) contributed to over 4.14 million deaths globally in 2019, twice as much as the numbers in 1990 (2.04 million deaths). In Europe, estimates from the European Environment Agency point to 307,000 annual premature deaths linked to fine PMs alone (PM2.5, i.e. particles smaller than 2.5 micrometres in diameter), a death toll that is one order of magnitude larger than the one of the other major pollutants, e.g. 40,400 annual deaths for nitrogen dioxide and 16,800 for ozone (European Environment Agency, 2021). This problem highlights the importance of accurately describing the spatiotemporal distribution of air pollutants, in particular PM. However, building and maintaining an extensive network of ground-level monitoring stations is expensive and not sustainable in many countries (Maag et al., 2018), which limits our capacity to derive spatially-representative estimates of the main air pollutants for epidemiological modelling and health impact assessment (Zhang et al., 2021).

Compared with surface PM (surface level aerosol particles), the total column of atmospheric aerosols can be monitored over a larger geographical coverage thanks to satellite observations (Griffin, 2013). The total column of atmospheric aerosols is generally measured as Aerosol Optical Depth (AOD), by detecting how much sunlight is absorbed or scattered by suspended particles and surrounding gases. In general, higher AOD values indicate more aerosols in the atmosphere, which can be associated with higher levels of PM. Therefore, AOD is the common and important indicator for surface PMs estimations, especially in those locations without available PMs observations.

However, satellite-derived AOD data still present some downsides, which may hinder the use of AOD to estimate PM. The first downside is the large data gap in satellite AOD. Over 85% of satellite AOD measurements are missing globally (Kahn et al., 2009; Chen et al., 2019b), mainly due to cloudiness, surface reflectivity and low sun angle at high-latitudes (Wei et al., 2018; Gupta et al., 2016; Chen et al., 2019b). Secondly, the accuracy of satellite AOD is still subject to various factors, like instrument calibration, cloud contamination, and climate or geographic conditions (He et al., 2021). These uncertainties may lead that the correlation between satellite AOD and PM levels varies a lot in different locations, and the correlation in some regions is always lower. For example, the Pearson correlation coefficient (PCC) between satellite AOD and PM levels in Europe and South America were found to be the lowest (from 0.1 to 0.12) by Christopher and Gupta (2020), against 0.45-0.70 in Northern America or East Asia. Lastly, polar satellites can only scan the surface a few times every day in each location, so space-based AOD measurements cannot fully match the continuous 24-hour ground level measurements. To solve this problem, geostationary satellites in some regions were launched recently but only provide data in recent years.

On the other side, aerosol reanalyses nowadays offer global gridded AOD estimates with no missing values, based on the assimilation of satellite AOD.  However, their resolution exhibit relatively coarse (of the order of 50-100 km) due to large computational requirements for fine-scale assimilation (Bouttier, 2009), and their products remain with some unavoidable bias mainly caused by uncertainties in the emission inventories (Huang et al., 2021), and some uncertainties from satellite or meteorological products that are assimilated.

In addition to these problems, the type and size distribution of aerosols can also change the amount of light that is scattered or absorbed by the aerosols, and therefore affect the AOD measurements and their relationship with PMs (Yan et al., 2017; Zang et al., 2021). For example, in regions with high levels of coarse particles in the atmosphere, those large particles such as dust can scatter more light than smaller particles, leading to the higher measurement of AOD. At this moment, the high level of AOD mainly represent those coarser particles rather than small particles like PM2.5. However, the size-resolved AOD information (fAOD and cAOD) are generally not taken into consideration for the estimation of PM. Due to the lack of reliable sources for



these components, previous studies have generally used total AOD as the default predictor of PM2.5 or PM10 (Ferrero et al.,
2019; You et al., 2015), even though more accurate models could be calibrated with fAOD and/or cAOD. Importantly, obtaining
reliable high-resolution data of the components of AOD still represents a major challenge (Yan et al., 2022). For the satellite
products, the MODIS fine mode fraction products are only available over oceanic areas, and its products over land have high
uncertainties (Levy et al., 2013); while the products from Polarization and Directionality of the Earth's Reflectance (POLDER)
is only available over short periods of time (Dubovik et al., 2019). To solve this problem, a few experimental studies (Chen et
al., 2020; Yan et al., 2022) attempted to improve satellite data quality by developing lookup table spectral deconvolution
algorithm (LUT SDA) and machine learning, but some limitations still persist, including the coarse spatial resolution (1 degree)
and the large data gaps.

Given the limitation of existing aerosol products, as well as the poor correlation between satellite AOD and PM2.5 in Europe,
we developed a new set of AOD models, and provided a 0.1 degree resolution daily AOD data set over Europe for the period
2003-2020 to better understand the spatiotemporal distribution of aerosols in the continent. First, based on Quantile Machine
Learning (QML) models fed with ground-level AERONET data and climate and aerosol reanalyses, we develop a high-resolution
daily dataset of AOD, fAOD and cAOD covering Europe over the period 2003-2020. Second, we investigate if these improved
AOD products provide a stronger correlation with surface PM. Compared with previous products, these QML estimates (AOD,
fAOD and cAOD) respectively shows higher correlation with PM10, PM2.5 and $PM_{coarse}$ (i.e. difference between PM10 minus
PM2.5), providing the foundation for indicating air pollutants distribution when ground-level stations are not available. Our
study was motivated by the need to fill the gap of missing satellite aerosol information, but also to provide reliable fine-mode
and coarse-mode aerosol estimates. At a longer-term horizon, it is aimed at providing new spatiotemporal PM exposure estimates
that could be used in epidemiological studies or for environmental surveillance.
**2. Data**
**2.1 AERONET data**
We collected cloud-screened, ground-based AOD data from AERONET v2.0 (Holben et al., 2001). This data source also
provides the decomposition of total AOD into fAOD and cAOD, based on a Spectral Deconvolution Algorithm validated by
O'neill et al. (2003). As it is commonly done in the literature, AERONET data are here considered as the "ground truth" to
validate other aerosol data (Gueymard and Yang, 2020; Bright and Gueymard, 2019). Our data are from 257 sites located in the
research domain here considered: 27-72N x 25W-45E. To be comparable with the satellite and reanalysis data, the AERONET
AOD data at 550 nm was interpolated from the two nearest wavelengths, i.e. 500 nm and 675 nm (Gupta et al., 2020; Duarte and
Duarte, 2020).
**2.2 Satellite data**
We collected daily AOD data for the period 2003-2020 from MODIS v6.1(https://ladsweb.modaps.eosdis.nasa.gov/), which is
based on the Multiangle Implementation of Atmospheric Correction (MAIAC) algorithm. The data are available at a 1km x 1km
spatial resolution over Europe. MAIAC uses time series analysis and an image-based processing algorithm to provide accurate,
fixed-grid aerosol estimations (Lyapustin et al., 2011). We further filtered the MAIAC data according to the quality assurance
flags of the NASA guidelines (i.e. Quality Assurance of cloud mask = clear sky)(Lyapustin et al., 2018).





**2.3 MERRA-2 reanalysis**


We also retrieved data from MERRA-2, the atmospheric composition reanalysis developed by NASA. To develop this product,
the Goddard Earth Observing System Model (GEOS-5) data assimilation system was used to ingest multiples sources (e.g. data
from AERONET sites, MODIS, MISR and AVHRR sensors (Randles et al., 2017)) to simulate aerosol data with the Goddard
Chemistry, Aerosol, Radiation, and Transport (GOCART) model. MERRA-2 provides assimilated 3-hourly aerosol data at a
resolution of 0.625°×0.5° from 1980. Previous studies (Che et al., 2019) showed that MERRA-2 reproduces the general trends
in annual and seasonal AOD, both at regional and global scales, but with significant biases in some locations.
**2.4 CAMS reanalysis**
Compared with MERRA-2, CAMSRA, the air quality reanalysis from the European Centre for Medium-Range Weather
Forecasts (ECMWF) (Inness et al., 2019), assimilates the hourly AOD data from Envisat's AATSR sensor, NASA's MODIS
Aqua and Terra sensors since 2003 and in-situ measurements from a wide range of sources (Bozzo et al., 2017; Flemming et al.,
2015), although, its spatial resolution is relatively coarser (0.75° x 0.75°). Previous studies showed that the estimates of AOD in
CAMSRA are slightly poorer than in MERRA-2 (Gueymard and Yang, 2020).
**2.5 ERA5 reanalysis for meteorological data**
Previous studies (Huang et al., 2007; Zhou and Savijärvi, 2014; Tai et al., 2010; Gui et al., 2019; Yan et al., 2022) have analysed
the associations between weather conditions and the concentration of fine- and coarse-mode aerosols. For example, high-pressure
events, characterised by atmospheric stability and low winds, retain the smaller particles, which is seen with higher-than-normal
fine-mode aerosol levels (Tai et al., 2010; Gui et al., 2019). Moreover, rainfall washes out the particles from the lower part of
the troposphere, especially the largest particles. There are other pathways by which aerosols can also affect weather conditions,
for example by reflecting and absorbing the incoming UV radiation (Zhou and Savijärvi, 2014), or by changing the conditions
for the condensation of water in the cloud(Huang et al., 2007). We therefore collected data from several atmospheric, oceanic
and land surface variables, such as boundary layer height, downward UV radiation, cloud cover, surface pressure and
precipitation, from ECMWF ERA-5 reanalysis, which provides data since 1950 at a resolution of 0.25º x 0.25º.
**2.6 ERA5 reanalysis for land surface data**
Apart from meteorological data, the land surface data also has important impacts on aerosol. As forests contribute to a large
extent to particle removal, previous studies found the deposition velocity of ultrafine particles is generally more sensitive to leaf
area index than leaf area density (Lin et al., 2018; Huang et al., 2015). Also, the dry deposition of particles is affected by
properties of the vegetation elements (such as leaves and branches) and soil types (Grönholm et al., 2009). Thus, we found the
significant contributions of leaf area index high vegetation, leaf area index low vegetation and soil types to aerosol. Higher Leaf
area index high vegetation means more evergreen trees, deciduous trees or forest, while Higher Leaf area index low vegetation
represents more crops and mixed farming, grass or shrubs. For bare ground or places with no leaves, both of them will be close
to zero. The soil types describe how coarse the soil is, representing the water holding ability of soil. Coarser soil generally has
lower water holding ability. Furthermore, land surface information is also important to surface reflectance, which further affects
the quality of satellite data included in reanalysis data.
**3. Methodology**
**3.1 Overall model**
In this study, we used quantile lightGBM (Light Gradient Boosting Machine) models to separately obtain predictions of AOD,
fAOD and cAOD. This gradient boosting framework is a high-performance, tree-based model requiring less computational time



than Gradient Boosting models or Random forest (Ke et al., 2017). The model also provides the intervals of the quantile
predictions to assess their uncertainty.

**3.2 Variable selection**

Our model is developed according to the methodological steps summarised in Figure 1. We first bilinearly interpolated all gridded
data to a horizontal resolution of 0.1° x 0.1° (i.e. around 9 km at mid-latitudes), and then extracted the corresponding values at
the longitudes and latitudes of the AERONET sites. To determine the variables included in the models, we conducted the Boruta
feature selection procedure (Kursa and Rudnicki, 2010) separately to AOD, fAOD and cAOD. The Boruta method is the robust
and powerful tree-based algorithm, which was successfully validated in previous simulation studies (Degenhardt et al., 2019).
For each variable, the method first generates a new shadow variable by randomly permutating the values of original variables,
and removes the original variable if there is no significant difference between the contribution to the model of the shadow and
the original variables. To reduce computational time and guarantee the stability, we conducted the Boruta method five times with
random subsamples of 20% of the sites to select those variables that are statistically significant (p-value < 0.05) (selected
variables are listed in Table S1).

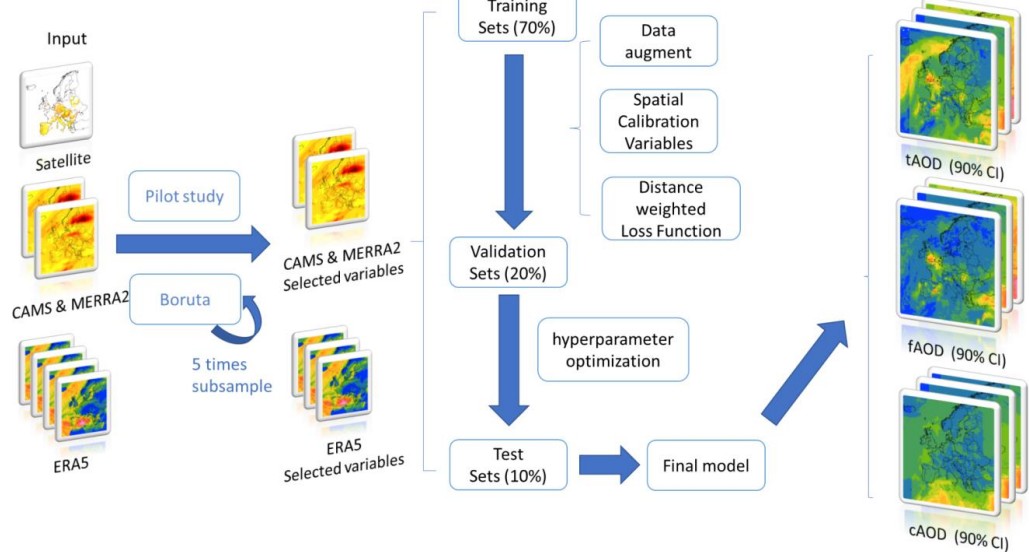


**Figure 1.** The workflow of Quantile machine learning (QML) for AOD, fAOD and cAOD

**3.3 Techniques to improve the models**

To improve the models, we randomly selected 70% of the sites as training data for the quantile lightGBM models. An additional
20% of the sites was used to optimize the model hyperparameter or to validate the following improving performance techniques
(see next paragraph). The remaining 10% of the sites was used to evaluate the out-of-sample predictive ability of the model, and
to select the optimal model configuration that was used to predict the daily AOD, fAOD and cAOD spatiotemporal estimates.
We applied three techniques to improve the accuracy and stability of the models:



- distance weighted loss function: given the heterogeneous spatial distribution of the 257 AERONET sites, the model is likely to be overfitted in those regions with a higher density of stations. To reduce the effect of this selective overfitting, we used distance weight factors in the loss function to decrease the weight in the regions with a higher density of stations. The mathematical formulation is

$$W_i = \frac{(D_i - D_{min})}{(\overline{D} - D_{min})} \tag{1}$$

$$L(Y, Y^*) = \sum_{i=1}^{n} W_i * L(y_i, y_i^*) \tag{2}$$

*where* $D_i$ *is* the distance of station *i* to its nearest site; $D_{min}$ and $\overline{D}$ denote the minimum and average distance to their nearest site; and $L(Y, Y^*)$ and $L(y_i, y_i^*)$ are the overall loss function (Y and Y* are the observation and prediction vectors) and the loss functions for station *i* ($y_i$ and $y_i^*$ are the observations and predictions at station *i*).

- minimum directional distance: we used this technique to analyse how predictions are affected by the distribution of sites along the longitudinal and latitudinal axes. Figure S1 illustrates how we constructed these spatial calibration variables. We first drew, for each station, the longitudinal and latitudinal axes to obtain the four quadrants. We defined a distance of 2000 km as an upper threshold in each direction. We then calculated the distance to the closest station in each direction, with a maximum value of 2000 km for the largest values. These distances in four directions were finally included as additional inputs in the models to account for the minimum directional distance.

- white-noise data augment: this technique was used to reduce the overfit of the model. It duplicates the original training datasets but adds white noise to the independent predictors. White noise is here defined by a Gaussian distribution with zero mean and the variance of corresponding independent predictors.

Actually, most of these techniques tend to increase the contribution of those sites are most distant from its nearby sites. To justify it, we used those 20% sites to validate the performance of these techniques. Also, we selected the sites with distance to their nearest sites above mean (463km), to examine the improvement for those locations with fewer stations. These sensitivity analyses (Table S3) show that the three techniques combined improved the quality of AOD, fAOD and cAOD predictions by around 38%, 27% and 44% in the regions with sparsely distributed stations.

Lastly, we also tested if the use of satellite MAIAC AOD improved the predictions (find details in the Section 3 of the Supplementary). Our analysis shows that there is no significant improvement in the final model after including satellite AOD, even after filling the missing values. This may be partially due to the fact that the reanalysis data already includes the satellite information (Bozzo et al., 2017; Flemming et al., 2015), and the satellite only accounts for a small fraction in the whole dataset. As a result, we excluded the MAIAC AOD data from our models.

**3.4 AOD Model Validation**

To evaluate the out-of-sample predictive capacity of the models, we separately obtained the spatial and temporal out-of-sample predictions by nested 5-fold cross-validation. For the spatial out-of-sample predictions, we randomly divided the 257 AERONET sites into five equal-sized subsamples. In each loop of generating predictions, we randomly used four subsamples for model training and tuning, and the other one for obtaining the out-of-sample predictions. In each model training, we further divided those four training subsamples to 80% for modelling, and the rest 20% for model hyperparameter optimization. This process was repeated for each of the five subsamples, generating five models and out-of-sample predictions for all the sites. For the temporal out-of-sample predictions an analogous strategy was used, but with six subperiods of three consecutive years, covering our 18-year period. Table S1 shows that the spatial and temporal models have similar structures, hyperparameters and results, indicating that our model strategy is stable to data training strategies.





To interpret these out-of-sample predictions, we compared these predictions with the AOD estimates from MERRA-2, CAMSRA
and satellite MAIAC data, by evaluating all of them against the AERONET data. For the fAOD and cAOD, as the reanalysis
data did not provide fine-mode and coarse-mode information, the satellite fAOD and cAOD is the only source for comparison.
Recently, Yan et al. (2022) have developed Phy-DL (PDL) Fine-mode Fraction (FMF) global products, and found their product
outperforms existing FMF products: Polarization and Directionality of the Earth's Reflectance (POLDER) FMF, Multi-angle
Imaging Spectro Radiometer (MISR) FMF and MODIS FMF. Their corresponding correlation with AERONET AOD are 0.78,
0.48, 0.42 and 0.37. Besides, the general correlation between PDL fAOD ( = PDL FMF * MAIAC AOD) and AERONET AOD
(PCC = 0.781) in our domain is similar with the result reported by Yan et al. (2022). Given that the PDL fAOD has been shown
to be an optimal method for global comparisons, we selected it as the primary reference for comparing the QML predictions.
The evaluation metrics, including R-squared, NMB (Normalized Mean Bias), NRMSE (Normalized Root Mean Square Error),
90% PI (predictive intervals) coverage, and the percentage of predictions within the expected error envelopes of 20%, are
described in Table S3.

**3.5 Exploratory correlation analysis with Surface PMs**

The future application of QML AOD, fAOD and cAOD predictions is to provide more information for predicting $PM_{10}$, $PM_{2.5}$
and $PM_{coarse}$ , thereby it is important to provide additional insights into the relationship between aerosol size distribution and
surface PMs. Once we obtain the reliable data for AOD, fAOD and cAOD, we can now undertake the investigation of their
relationship. Also, we can examine whether the QML products can potentially be a better indicator for PMs. First, we obtained
QML AOD prediction for locations with monitoring sites of PM, and then compared spearman correlations coefficient (SCC)
results among different AOD products with $PM_{10}$, $PM_{2.5}$ and $PM_{coarse}$. The potential monotonic relationship between AOD and
PM is generally not strictly linear, so SCC is a better evaluation tool than PCC. Notably, the location of ground-level PM sites
generally does not coincide with the AERONET sites, so correlation results are less likely to be affected by overfitting.

**4. Results and discussion**

**4.1 Description of AERONET Data**

Figure 2 shows the spatial and temporal distribution of AERONET station data. On average, each site has 662 observations
during the 2003-2020 period, with approximately 28% of sites having over 1,000 observations. Early-built sites (constructed
before 2010) are distributed relatively evenly throughout Europe, allowing models in the early period to be trained with data that
represents the continent as a whole. The annual sample size of observations has been increasing during the last two decades, and
reached its maximum in 2018. In relative terms, the average number of daily observations gradually increased until 2011, when
it stabilised at around 150 annual days per site. This plateau period (2011-2020) indicates that the average number of observable
days for most sites is restricted to around 150 days due to some external factors. For example, the sun photometer requires clear
skies (Holben et al., 2006; GLOBE 2010)). Sometimes, some stations require calibration and are usually taken offline for a few
months intermittently. Before the plateau, the increasing number of observable days per site indicates that some AERONET sites
may have not operated regularly in this earlier period, which would limit the performance of QML AOD models.

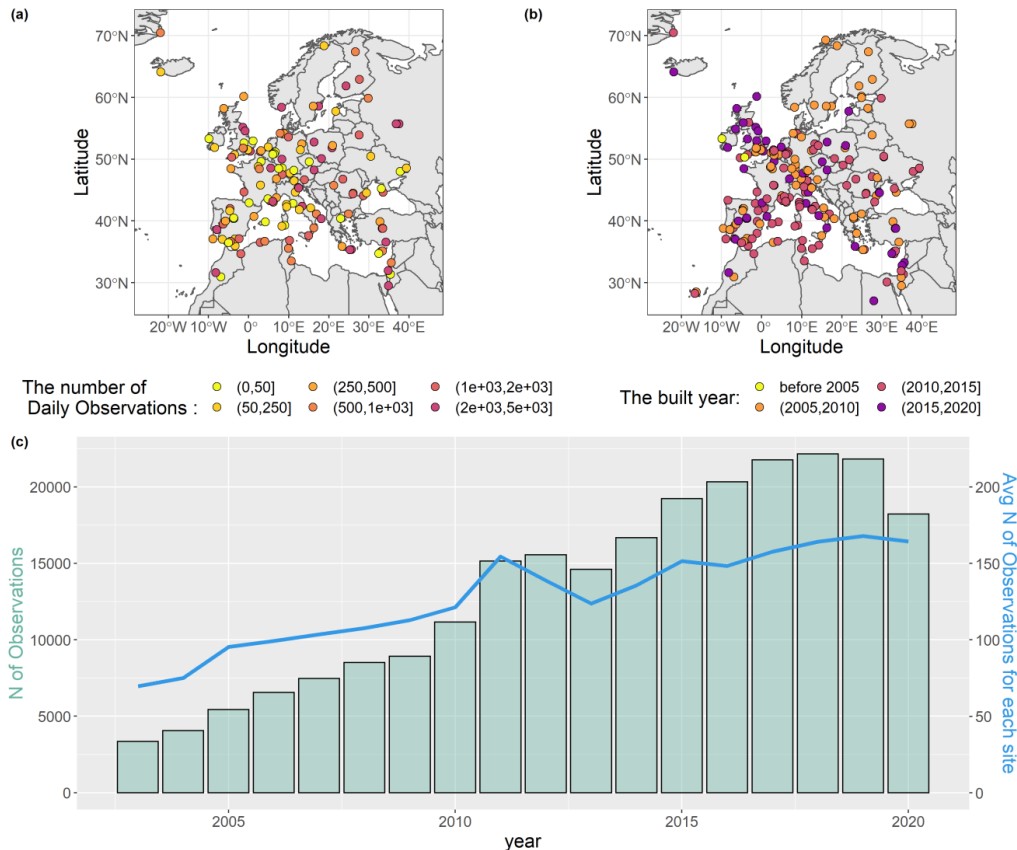

**Figure 2. S**patial and temporal distribution of AERONet sites: (a) number of daily values, (b) built-year and (c) number of observations for whole domain (left axis, green bars) and average of observable days per site (right axis, blue line).

Figure 3 depicts the spatiotemporal distribution of AERONET AOD, fAOD and cAOD observations. For instance, AOD and fAOD are higher in central and eastern Europe, while cAOD is higher in southern Europe. This indicates that central and eastern Europe are largely affected by anthropogenic pollution sources, mainly associated with anthropogenic (Bellouin et al., 2005) and secondary aerosols (e.g., sulphate, nitrate, ammonium) (Zhao et al., 2018; Seinfeld and Pandis, 1998), while southern Europe is influenced by intrusions of mineral dust from the Sahara desert, as well as the advection of sea salt from the Mediterranean Sea (Meloni et al., 2008; Prospero et al., 2014). Regarding the long-term trend in AOD, the annual values have decreased at a rate of 28.5% per decade for AOD and 27.2% for fAOD, while they have remained stable for cAOD. Given that fAOD is largely associated with anthropogenic emissions, its decreasing trend reflects the reduction in anthropogenic pollution resulting from air quality plans implemented in Europe.

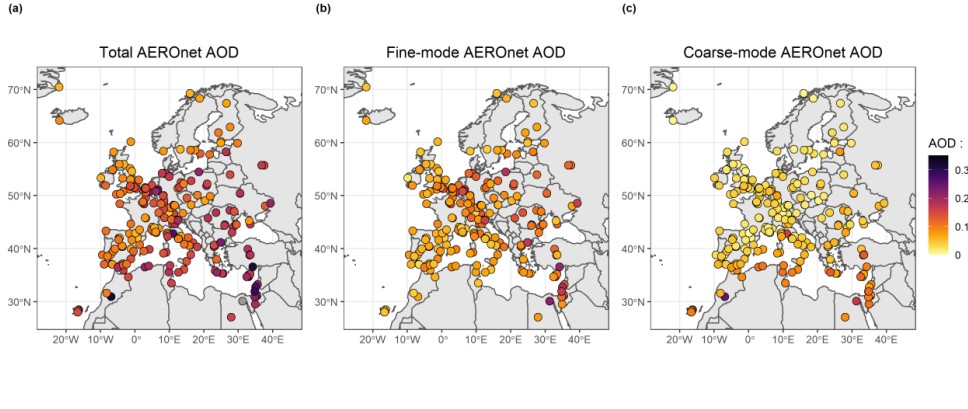

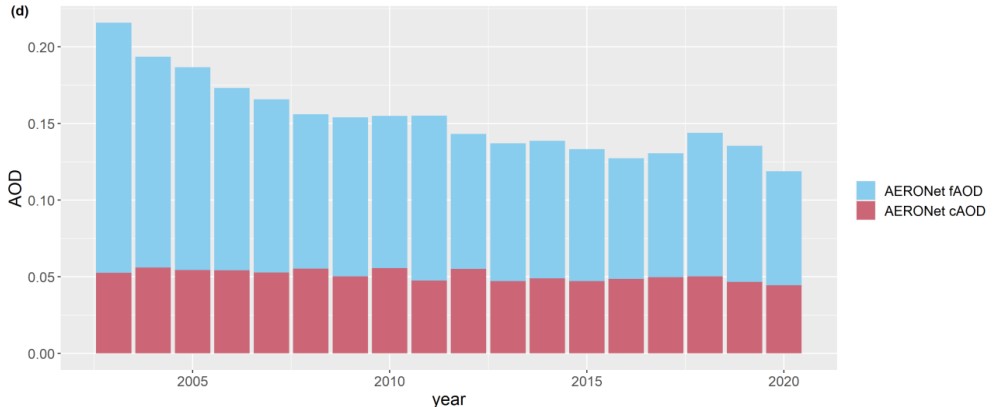

**Figure 3**. Spatial and temporal distribution of the median value of AERONet (a) AOD, (b) fAOD and (c) cAOD data. (d) Temporal evolution of AOD (red+blue), fAOD (blue) and cAOD (red).

## 4.2 Spatial and Temporal Evaluation

To further validate the AOD models, we compared in time and space the AOD predictions with the estimates from MERRA-2, CAMSRA and MAIAC. Given the large fraction of missing values in satellite MAIAC AOD (64%), we divided the validation results into two subgroups of dates and sites, i.e. those with ("Sat scenario") or without ("Non-Sat scenario") available satellite AOD data, in order to guarantee the comparability among the different products, and provide a better understanding of the factors limiting the performance of the models. Unlike MERRA-2, CAMSRA and MAIAC, we used our models to generate out-of-sample predictions in space and time in order to shed more light into the validation.

### 4.2.1 Total AOD product

Figure 4 validates the different AOD products against the AERONET observations, by comparing QML's spatial and temporal out-of-sample AOD predictions with MERRA-2, CAMSRA and MAIAC AOD. The NMB maps show that, in general, QML predictions agree best with AERONET data, but they are slightly underestimated both in the spatial and temporal out-of-sample predictions. Additionally, the QML also has the lowest NRMSE, approximately 33% smaller than the other products. In contrast,

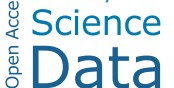

CAMSRA is the product with the largest overestimation, especially in Northern Europe, United Kingdom and some
Mediterranean coastal areas. Meanwhile, satellite AOD tends to underestimate inland areas and overestimate coastal areas.
In Figure 4 (a1-e1), the box plots of the bias (estimation minus AERONET) are displayed in each decile (Q1-Q10) of the
AERONET AOD, with the range in each decile additionally shown in Table S4. The orange and green lines represent the
expected error (EE) envelopes of ±40 % and ±20 % for each decile according to the methodology described in previous studies
(Levy et al., 2010; Xiao et al., 2016; Yan et al., 2022)). Overall, QML predictions show a high agreement with AERONET data,
with approximately 91% (temporal out-of-sample predictions) and 89% (spatial) of the predictions falling within 20% expected
error (EE) envelopes. In contrast, the 20% EE in the other products ranges between 76% and 78%. QML also exhibits the highest
R-squared, i.e. 0.71 and 0.68 for the temporal and spatial out-of-sample predictions.
In the Sat scenario, all the products tend to overestimate the AOD in the lower AOD deciles, and to underestimate it in the higher
deciles. However, QML has the highest proportion of predictions within 20% EE (i.e. over 93%), especially with less bias in the
lower deciles. In the Non-Sat scenario, all products perform relatively worse compared to the Sat scenario. This is partially due
to the fact that reanalysis data assimilates satellite data. QML models also benefit from including reanalysis data as input.
CAMSRA AOD is the product with stronger overestimations in almost all deciles (Q1 to Q8), while MERRA has similar
performance as CAMSRA from Q1 to Q6. In comparison, QML mainly narrows the bias among those quantiles (Q1 to Q8), and
its rate of falling within 20% EE reaches 86%-89%.
Overall, both scenarios show that QML AOD mainly provide better predictions in lower quantiles, where other products tend to
overestimate AOD. This implies that QML AOD is less likely to overestimate values in the lower range, but like other products,
it may still exhibit some overestimates in higher ranges, leading to a slightly negative NMB.
To assess the performance over time, Figure 5 presents the annual performance of AOD products based on three criteria: R-
square, NMB, and NRMSE. QML AOD predictions consistently achieve higher R-squared values than reanalysis or satellite
estimates. Regarding NMB, all products show slightly increasing trends over the years, with MERRA-2 and MAIAC estimates
shifting from negative to positive values. This trend could be due to the decrease in AERONET AOD over time, coupled with
the tendency of most products to overestimate the AOD in the lower deciles. In contrast, the NMB in QML is relatively constant.
As for NRMSE, all products show slight decreases over time, with the lowest values in QML throughout the entire period.



**Figure 4.** The spatial distribution maps of normalize mean bias (NMB) (a-e) and box plots of AOD estimation bias (Estimation minus AERONet data) for different quantiles of AERONet AOD (a1-e1) in 2003-2020. These data sources include: MERRA-2 AOD (a-a1), CAMSRA AOD (b-b1), MAIAC AOD (c-c1), QML spatial out-of-sample prediction (QML AOD Spat) (d-d1) and QML temporal out-of-sample prediction (QML AOD Temp) (e-e1). The N(%) is the sample size (proportion); 20% and 40% EE are expected error envelopes with 0.05 ±20 % observation and 0.05 ±40 % observation. The upper, middle, and lower lines in each box are the 75th, median, and 25th percentiles, respectively.

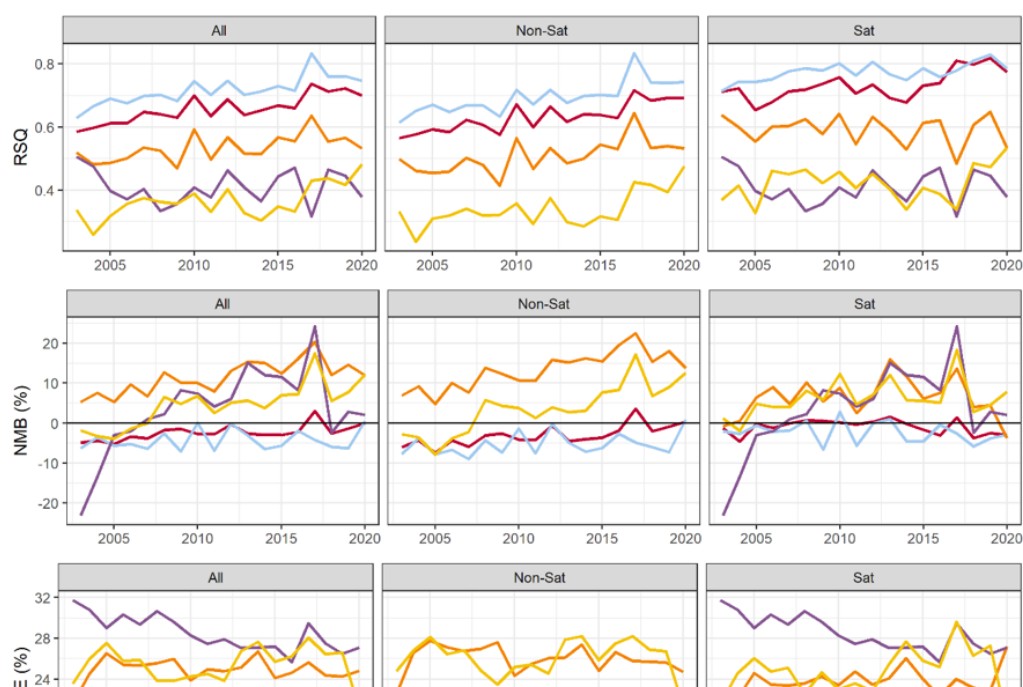

**Figure 5.** Comparison of the performance (R-squared, NMB, NRMSE) between different data sources: QML spatial out-of-sample prediction (QML AOD-Spat), temporal out-of-sample prediction (QML AOD-Temp), CAMS AOD, MAIACAOD, MERRA-2 AOD.

After splitting the data into two scenarios, it was observed that most products performed slightly worse in the Non-Sat scenario, while QML consistently maintained better performance in both scenarios. Additional validation statistics for calendar months (Fig S2) and days of the week (Fig S3) are provided in the Supplementary, which also show that the QML temporal and spatial predictions outperform the estimates from reanalysis and satellite AOD. All products fit better with AERONET data in the warm season (April-September), while it is more challenging to predict AOD in the cold season (October to March) due to cloud and rain, which reduces the amount of sun photometer observations (GLOBE 2010; Holben et al., 2006). The reanalysis data, particularly the CAMSRA AOD, are more likely to overestimate the AOD in the warm season and to underestimate it in the cold season, while QML performance remains relatively constant among seasons. Regarding the weekly cycle, there are no significant

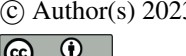



differences between the days of the week, possibly due to the relatively smaller influence of human activities on column-
integrated AOD.

QML provides 90% prediction intervals (PI) as an estimation of the uncertainty of the predictions, and its performance is shown
in Table S4. The average coverage of the QML temporal and spatial PI is 84.2% and 83.5%, respectively, slightly lower than the
expected value of 90%. QML still faces some difficulty in predicting intervals for the first and last deciles, with only a coverage
of 55% in Q1 and 68% coverage in Q10. This indicates that the intervals are slightly underestimated by our quantile models.

**3.2.2 Fine-mode AOD product**
Figure 6 validates the QML and PDL fAOD products against the AERONET observations. Generally, QML fAOD does not
exhibit any spatial pattern in its NMB, but with some underestimates in certain locations. Additionally, its overall NRMSE is
lower than that of PDL fAOD. In contrast, PDL fAOD exhibits an uneven pattern in its predictions, with overestimates in
Southwest Europe and underestimates in Central and Eastern Europe. The main difference between these two areas is their fAOD
concentration, with generally lower fAOD in Southwest Europe and higher fAOD in Central and Eastern Europe (as seen in
Figure 3 (b)). Thus, it can be inferred that this uneven pattern results from the preference of PDL to overestimate small fAOD
values and underestimate high fAOD values (as seen in Figure 6 (a1)). However, Compared to PDL fAOD (Fig. 6 (a1-c1)), QML
fAOD narrows the bias, especially in smaller deciles (Q1-Q5), so that over 86% (spatial) and 87% (temporal) of the predictions
fall within 20% expected error (EE) in Sat scenarios. In Non-Sat scenarios, QML still maintains 77-79% of the predictions within
20% EE. The overall R-squared of QML reaches 0.66 (spatial) and 0.68 (temporal).

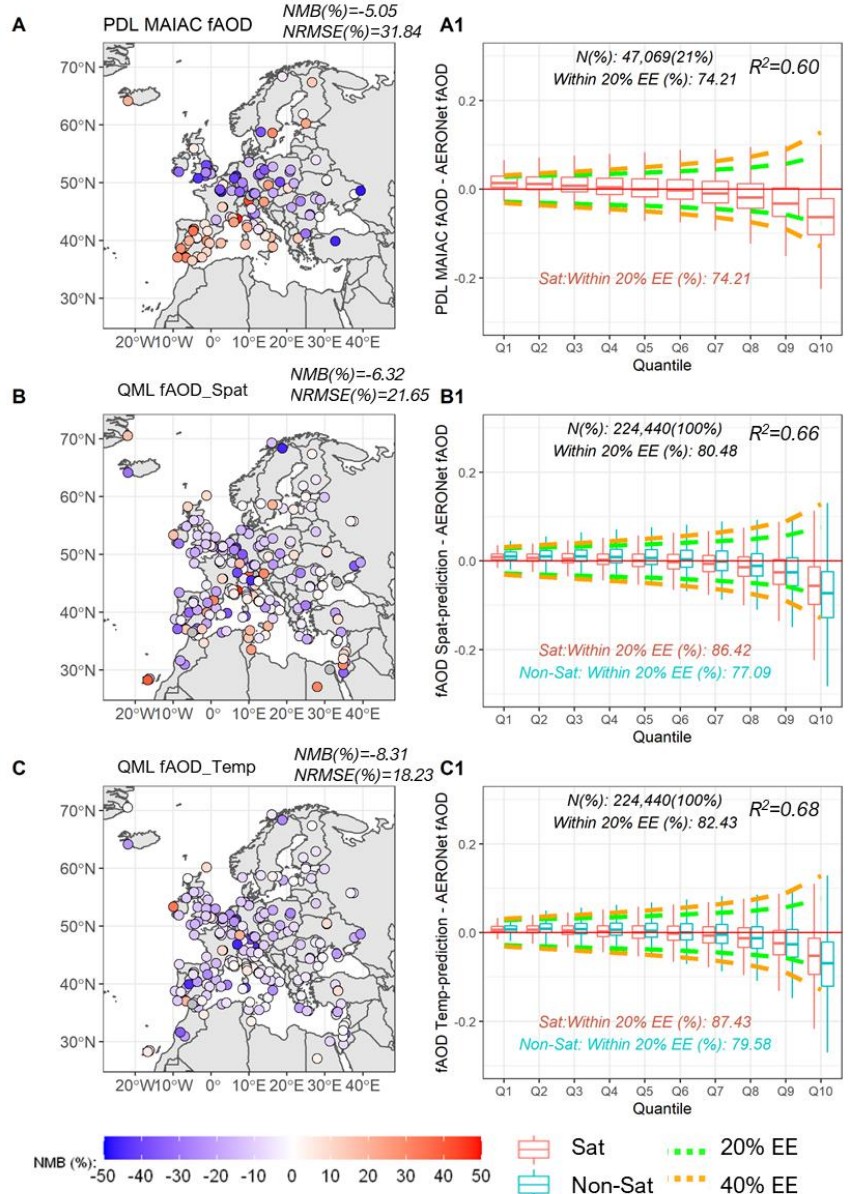

**Figure 6.** Spatial distribution maps of normalize mean bias (NMB) (a-c) and quantile box plots for for different quantiles of AERONet fAOD (a1-c1) in 2003-2020, including: Phy-DL Satellite fAOD (PDL MAIAC fAOD) (a-a1), QML spatial out-of-sample prediction (QML fAOD-Spat)(b-b1) and temporal out-of-sample prediction (QML fAOD-Temp) (c-c1). The N(%) is the sample size (proportion); 20% and 40% EE are expected error envelopes with 0.025 ±20 % observation and 0.025 ±40 % observation. The upper, middle, and lower lines in each box are the 75th, median, and 25th percentiles, respectively.



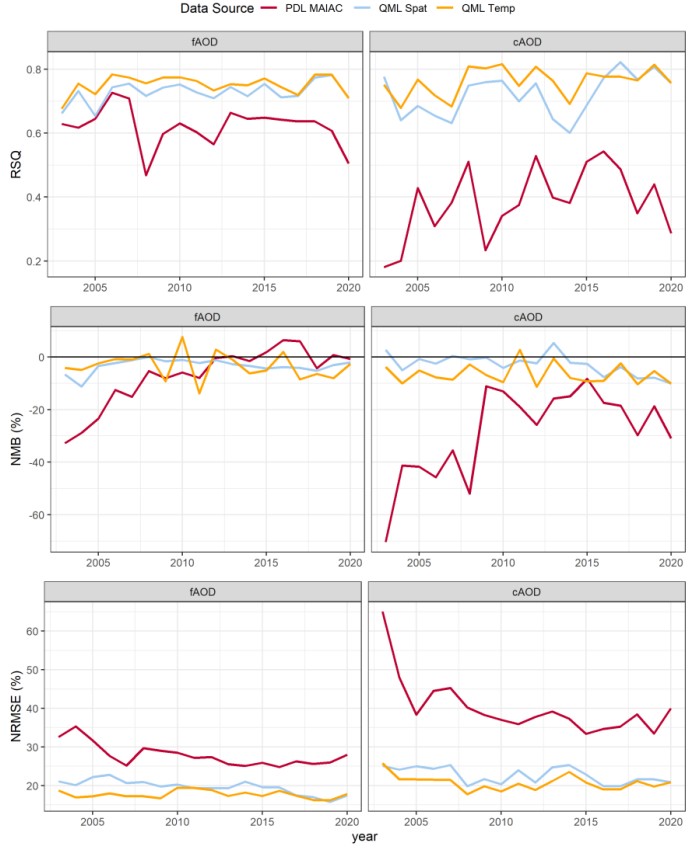

354

**Figure 7.** The annual performance (R-squared, NMB, NRMSE) comparison between different data sources in the satellite scenario: QML spatial out-of-sample prediction (QML fAOD Spat-Prediction), temporal out-of-sample prediction (QML fAOD Temp-Prediction), Phy-DL Satellite fAOD(PDL MAIAC fAOD).

As the PDL fAOD is only available in the satellite scenario, Figure 7 (left columns) compares the year-to-year performance between PDL and QML fAOD in the Sat scenario. The R-squared values indicate that the performance of PDL fAOD exhibits greater variability over time than QML, and the difference between the two products appears to increase slightly over years. The NMB (%) shows a strong underestimation of PDL fAOD before 2008, partly because of its uneven spatial bias pattern during that early period (Fig 6A). Prior to 2008, more AERONET sites were located in western Europe (Fig. S4), where PDL fAOD was underestimated. Afterward, more sites were set in areas in which the product tends to overestimate the predictions (e.g., Southwest Europe), offsetting the bias and bringing the NMB of PDL fAOD closer to 0 after 2008. The NRMSE results also suggest that PDL fAOD has a larger error than QML fAOD. In contrast, the QML predictions are generally better in terms of NRMSE, and errors remain stable over the years.

We note that the availability of PDL AOD in our dataset is lower than the satellite MAIAC AOD (21% VS 36%), because PDL FMF is not always available when satellite AOD is available. Figure S5 shows that the R-squared of QML fAOD in the Non-Sat scenario is about 10% lower than its performance in the Sat scenario, and its NMB also indicates a stronger underestimation than in the Sat scenario.


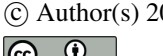



In the seasonal analysis (Fig. S6), all fAOD products show a similar pattern to total AOD, and perform better in summer. In the
Sat scenario, QML fAOD provides better and more stable performance than PDL fAOD. At the weekly level, there is no
difference in performance among different days of the week (Fig. S7). The performance of PI in QML is shown in Table S5,
with the average coverage for QML temporal and spatial PI at 83.3% and 84.4%, respectively, which is similar to the results of
total AOD.

**3.2.3 Coarse-mode AOD product**
Figure 8 shows the evaluation of QML and PDL cAOD products against AERONET observations. All cAOD products have
overall negative NMB, but the NMB of QML is around half that of PDL cAOD, due to the relatively strong underestimation of
PDL cAOD in some locations of Central Europe. The overall NRMSE in QML is also around 18-19% lower. Around 91%(spatial)
and 93%(temporal) QML predictions fall within the 20% EE in the Sat scenario (Fig.8 (a1-c1)), while 83% of the PDL predictions
are within the 20% EE. Moreover, the R-squared difference is larger, i.e. 0.65-0.67 in QML and 0.38 in PDL. For the non-sat
scenario, 87% (spatial) and 89%(temporal) of QML predictions fall within 20% EE, around 4% less than in the Sat scenario.

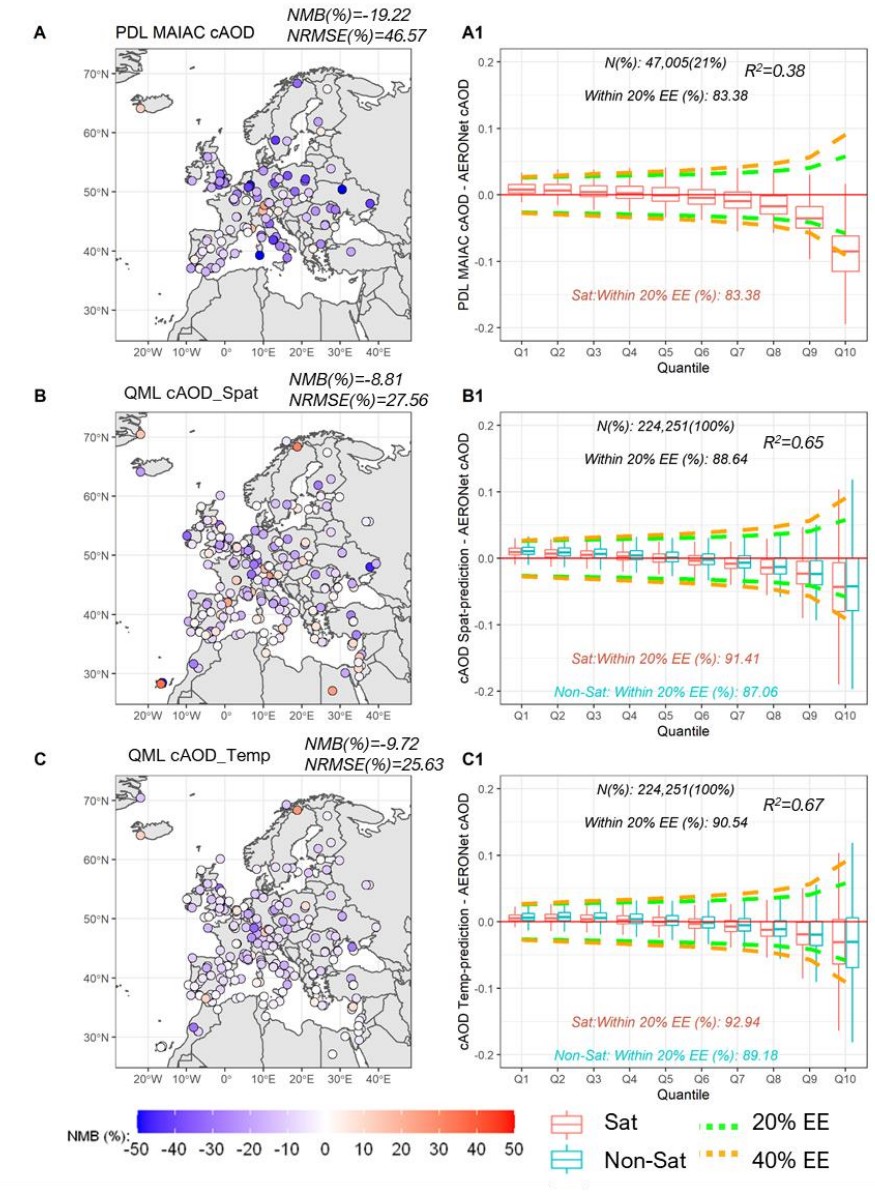

**Figure 8.** the spatial distribution maps of normalize mean bias (NMB) (a-c) and quantile box plots for for different quantiles of AERONet cAOD (a1-c1) in 2003-2020, including: Phy-DL Satellite cAOD(PDL MAIAC cAOD) (a-a1), our spatial out-of-sample prediction (QML cAOD Spat-Prediction) (b-b1) and temporal out-of-sample prediction (QML cAOD Temp-Prediction)(c-c1). The N(%) is the sample size (proportion); 20% and 40% EE are expected error envelopes with 0.025 ±20 % observation and 0.025 ±40 % observation. The upper, middle, and lower lines in each box are the 75th, median, and 25th percentiles, respectively.

Figure 7 also shows the comparison of the year-to-year performance for between PDL cAOD and QML cAOD, when both of them are available. We can see the performance difference between QML cAOD and QML fAOD is smaller than the differences between PDL products. The R-squared of PDL cAOD is relatively lower and more volatile than PDL fAOD, while QML cAOD



maintains a similar performance as QML fAOD. And the PDL tends to underestimate cAOD to a lower extent over years, while
it exhibits stronger bias in its NRMSE. Since PDL cAOD is calculated by subtracting PDL fAOD from Satellite MAIAC AOD,
the large differences in performance between PDL fAOD and PDL cAOD suggest that the modelling of PDL FMF might rely
too heavily on the calibration of AERONet fAOD, while neglecting the contribution from cAOD data. We therefore conclude
that it is important to be cautious when using PDL FMF to extract cAOD values, while the QML appears to resolve this potential
bias in cAOD prediction.

The result of the non-satellite and all scenarios of QML cAOD is displayed in Figure S8. This figure shows that the difference
in the performance between the Non-sat and sat scenarios is smaller compared with QML fAOD. The reason of these more
accurate predictions for QML cAOD in the Non-sat scenario is that, compared with fAOD, the cAOD is more influenced by
larger-scale sources (e.g., dust storms or sea salt advection), which means that their spatial distribution is generally more
homogeneous. Thus, it is easier to predict even when satellite information is absent, while fine aerosols come from various
anthropogenic sources.

When the analysis is performed by month, the QML cAOD product consistently outperforms the PDL cAOD, while the latter
exhibits larger variability across months (Figure S9 and S10). The performance of PI in QML can be found in Table S6, and the
average coverage of the QML temporal and spatial PI is 81.8% and 79.2%, respectively.

**3.3 Correlation between AOD products and particulate matter components**

After this in-depth validation of the three AOD products, we here investigate the Spearman correlation between AOD, fAOD
and cAOD with PM2.5, PM10 and PMcoarse, to explore the potential optimal indicator for surface PMs. To ensure comparability
of analyses, we only computed the correlation for locations with at least 100 AOD-PM data pairs.

Figure 9 displays the fishnet maps of SCC between ground-level PM2.5 and different AOD products. The fishnet maps show the
show the mean correlation for stations over a 0.1°x0.1° grid, given that the network of over 2,600 PM2.5 sites are too dense to
show in a point map. We excluded cAOD results from these comparison maps because the resulting correlations were not
statistically significant (Table S7). Among the other AOD products, the strongest correlation with daily ground-level PM2.5 was
found in QML fAOD (0.45) and QML AOD (0.40), followed by PDL fAOD (0.29), MERRA-2 (0.18), MAIAC (0.17), and
CAMSRA AOD (0.10). The correlation between PM2.5 and PDL fAOD is higher than with MAIAC AOD, which is consistent
with previous research. Furthermore, QML fAOD performs better than QML AOD, indicating that the fine-mode component of
AOD may be a better proxy for PM2.5 (Lin et al., 2019; Zang et al., 2021; Yan et al., 2017). Both QML AOD and QML fAOD
have a stronger correlation with PM2.5 than PDL fAOD, while they exhibit lower NRMSE to AERONet data (21.65 (QML
fAOD), 21.25 (QML AOD), and 31.84 (PDL fAOD)). Regarding the spatial pattern, the strongest correlations with PM2.5 across
most products are found in Western Europe. One possible explanation for this phenomenon is that the aerosol in Western Europe
has a higher proportion of anthropogenic aerosol sources. These sources are more likely to concentrated in lower atmosphere or
in smaller areas, such as urban centres, which may contribute to the stronger relationship between AOD and PM observed in this
region.





**Figure 9.** Spearman correlation fishnet maps between PM2.5 and different AOD products: CAMS AOD (a), MAIAC AOD(b), QML total AOD predictions (QML AOD) (c), MERRA-2 AOD (d), Phy-DL Satellite fAOD (PDL fAOD) (e) and QML fine-mode AOD predictions (QML fAOD)(f).





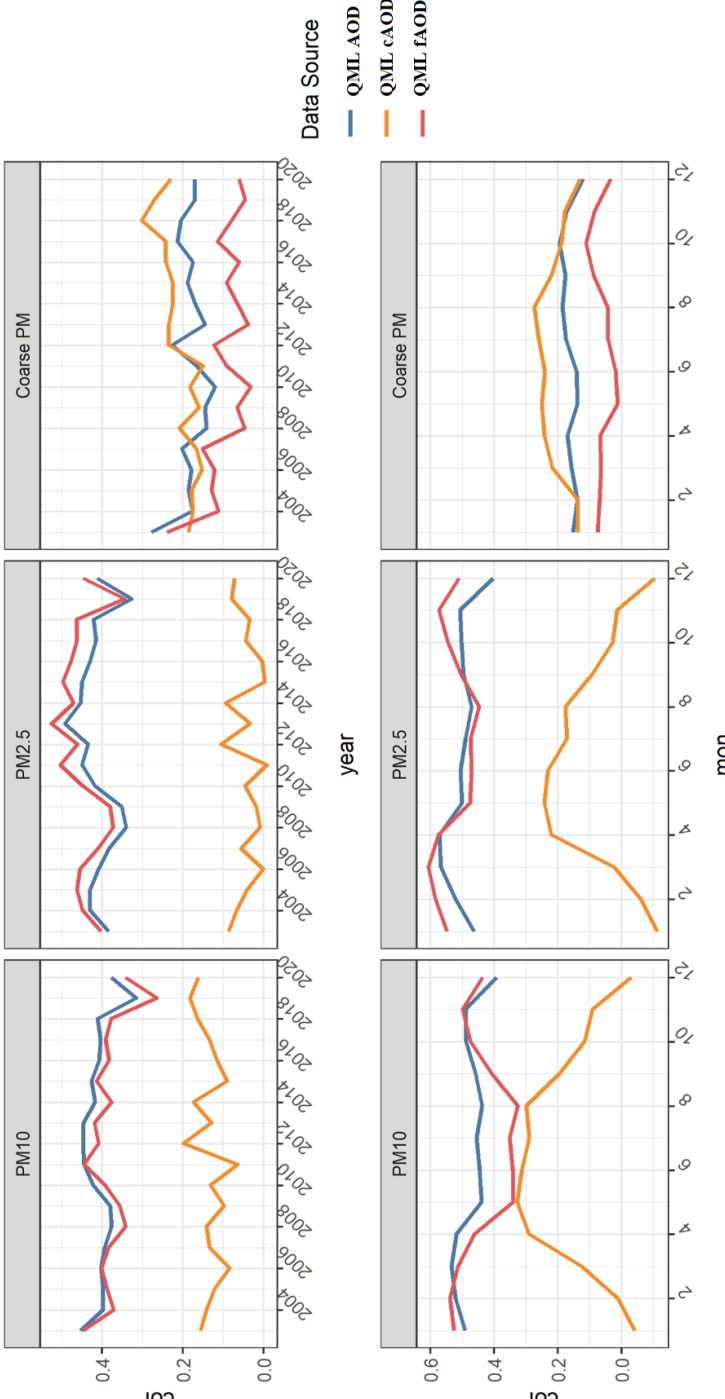

**Figure 10.** Spearman correlation between PM10, PM2.5, PMcoarse and three AOD products: QML AOD, QML cAOD and QML fAOD.





The correlation analysis with PM10 and PMcoarse is additionally shown in Fig. S11 and Fig. S12, respectively. The spatial
pattern of the correlations with PM10 is similar to the one obtained with PM2.5, but the SCC between PM10 and QML AOD is
higher than the one with QML fAOD, around 0.41 and 0.37, repetively. For the Coarse PM, the correlation with QML cAOD
is the strongest, reaching 0.26, especially in the southern Europe.

As the correlation with different sizes of PM is consistently higher in QML products (Table S7), we explored the relationship
between three QML AOD components and different-size PMs. Figure 10 presents the correlation analysis among QML AOD
products, PM10, PM2.5, and PMcoarse in locations with at least 100 AOD-PM pairs during 2003-2020. The correlation in
different years suggests that AOD, fAOD and cAOD are the best candidates for PM10, PM2.5 and Coarse PM prediction,
respectively.

The correlation of QML fAOD with PM2.5 and PM10 is always highest in winter and lowest in summer (Fig. 10, second rows).
In contrast, the correlation of cAOD with PM2.5 and PM10 shows the opposite seasonal pattern: weaker or even negative in
winter, but stronger in spring and summer. In winter, the proportion of suspended fine particles increases due to more energy
consumption, such as domestic heating (Martins and da Graça, 2017). At the same time, the cAOD in winter is more contributed
by sea-salt aerosols in Europe (Zhao et al., 2018), with the help of the windy season under low pressure systems over the Atlantic
Ocean (Manders et al., 2009). Thus, we can observe the non-significant or even negative correlation between cAOD and PM2.5
in the winter. In spring and summer, when the Saharan dust transport is active (Meloni et al., 2008; Prospero et al., 2014), the
cAOD starts to have significant correlation with all particulate matter fractions, especially with the coarse. This also reduces the
contribution of fAOD to PMs in summer, especially to PM10.

As AOD, fAOD and cAOD are the best potential candidates for the prediction of PM10, PM2.5 and Coarse PM, respectively,
we further compare QML products with other products in AOD, fAOD and cAOD (Fig. S13). Among all types of AOD products,
the QML products always provide better indicators for ground-level particulate matter.

**3.4 Spatial distribution and trend analysis for QML AOD products**
In this section, we compared the spatial distribution of 18-year average maps from different data sources (Fig. 11) and their
trends over the study period (Fig. 12 and Fig. S14). The 18-year average of reanalysis products and QML products generally
have a similar spatial pattern at 0.1-degree resolution (Fig. 11), and show good agreement with the long-term average of
AERONET data. The Pearson correlation with AERONET data for the 18-year averages are: 0.89 (QML AOD), 0.85 (MERRA-
2), and 0.79 (CAMSRA). Also, the 18-year averages of QML fAOD and cAOD are highly correlated with corresponding
AERONET data (R=0.91 and 0.93). The uncertainty of QML products is shown as the relative predictive intervals width (RPIW,
the ratio of 90% predictive intervals width and corresponding estimates) in Figure S15.







**Figure 11.** 18-year averages (2003-2020) of Reanalysis AOD (CAMS and MERRA-2) (A1-A2), QML product (AOD, fAOD and cAOD) (B1-B3), Satellite AOD(MAIAC AOD, PDL fAOD and PDL cAOD) (C1-C3). The satellite plots only show values with more than 30% of available data due to large ration of missing values. Points represent the AOD values observed in AERONET.





**Figure 12.** Monthly time series of AOD averaged over the whole domain (a) and in those 46 AERONet sites with more than 10 years of data (b)




For the spatial pattern for aerosol, the high values of total QML AOD are mainly located in central and eastern Europe, the
Mediterranean and Northern Africa. However, the composition of these high-value AOD are different. For example, the high
fAOD are mainly located in inland areas of Europe, like the middle and eastern part of Europe (especially Northern Italy,
Southern Poland, Hungary, Serbia, Romania and Bulgaria). Meanwhile, the cAOD are mainly affecting Northern Africa, the
Mediterranean and some coastal areas in Europe. This composition pattern is consistent with the general pattern in AERONET
data, but gives more details in each exact location due to the extended coverage of the product. As the fAOD and cAOD mainly
come from different sources (anthropogenic vs dust/sea-salt sources) (Zhao et al., 2018; Seinfeld and Pandis, 1998; Yan et al.,
2022; Bellouin et al., 2005), the AOD in middle and eastern of Europe are more contributed by anthropogenic small-size aerosol,
while Mediterranean areas and Northern Africa are more impacted by Saharan dust.

Figure S14 shows the 18-year trend maps of QML (AOD, fAOD and cAOD), Satellite (MAIAC AOD, PDL fAOD and PDL
cAOD) and Reanalysis (CAMSRA and MERRA-2) products. To guarantee the robustness while calculating the annual trend,
we subset 46 AERONET sites (points in Fig. s14), with more than 10 years of data and at least 50 daily observations per year.
Generally, QML and CAMSRA have a good agreement with these 46 AERONET sites. All the total AOD products show
decreasing trends in Europe. Both total AOD and fAOD have stronger decreasing trends in central and Southeast Europe, while
the trend of the QML cAOD is not significant in most areas of Europe. As the fAOD is discriminates to some extent
anthropogenic aerosols (Bellouin et al., 2005), these trend results support previous findings (Crippa et al., 2016) in that the
decreasing aerosol emission in Europe are mainly driven by reduced anthropogenic emission (e.g., transportation and industrial
emission).

Figure 12(a) shows the monthly time-series plot of different AOD products in the Pan European domain. The four products show
similar monthly cycles: the values are higher in summer/spring, and lower in winter/autumns. The seasonal patterns are consistent
with previous findings (Chen et al., 2019a; Zhao et al., 2018). In summer, more secondary aerosols are formed under higher
insolation and temperature (Kulmala et al., 2014). Meanwhile, the more abundant water vapor in summer boosts the hygroscopic
growth of aerosols (Zheng et al., 2017). Even though the seasonal pattern is similar in all products, reanalysis data (CAMSRA
and MERRA-2) generally provide higher values in summer than others, and MERRA-2 values in winter are also higher.

To further confirm which estimation is closer to the ground-truth, Figure 12(b) further compares different products in those 46
AERONET sites. Compared with the whole domain (Fig. 12(a)), these 46-site data generally have a stronger decrease trend (Fig.
12(b)), because many of them are located in areas with steeper decreases (Fig s14), like central or western Europe. To evaluate
the trend consistency, we also calculate the absolute log ratio between the trend of each product and AERONET data, as the
Trend Inconsistency index (TI) (see more details in Table S3). If two trends are perfectly consistent, TI will be zero. Otherwise,
it will be far away from zero. The QML product keeps the better agreement with AERONET data (TI = 0.13), and the values
also fit closer to the ground-truth. The slope of CAMSRA is also close to the AERONET one (TI=0.22), but with a higher
intercept, due to its overestimation in summer. Lastly, the satellite and MERRA-2 AOD generally underestimate the decreasing
trend (TI= 0.44 and 0.86), but in a different way. The satellite mainly underestimates the AOD values before 2015, while
MERRA-2 overestimates in summer after 2015.

Lastly, we also plotted the trend of fAOD and cAOD grid data in Europe (Figure S16). The seasonal cycles and annual trends of
fAOD are similar to those of AOD, which indicates that the variation and trends of AOD in Europe are mainly dominated by
fAOD. The cAOD a relatively flat annual trend, and always peaks in spring and early summer (from April to June), consistent
with Saharan dust events in Europe being more frequent in spring and summer (Papayannis et al., 2008)



### 3.5 Summary of evaluation of the AOD products


To summarize the strength and weakness among different AOD products, Figure 13 summarizes the results in eight dimensions:
Accuracy, Stability, Percentage (%) of bias within 20% EE, Correlation with corresponding PMs, TI index, Coverage, Resolution
and Product Period. We only compared results in the common period 2003-2020 among different products.

For AOD, QMLproduct performs the best in these dimensions: it shows higher accuracy ($R^2$: 0.69 (QML AOD) vs 0.36-0.56
(others)) and stability (variation coefficients of $R^2$: 0.24 (QML AOD) vs 0.35-0.53 (others)), and it is a better indicator of ground-
level PM10 (Correlation with PM10: 0.41(QML AOD) VS 0.17-0.23(others)).

We also found out that the higher correlation with AERONET AOD is generally linked to a higher correlation with PMs, but not
always. For example, the CAMSRA is well fitted with AERONET AOD, but its correlation with PM10 is the lowest among the
four products. Some potential factors could explain this behaviour. First, the location of ground-level PM observations generally
does not coincide with the AERONET sites, and their distribution is denser and covers more areas in our domain. Therefore,
some unknown biases or factors from CAMSRA in those locations outside the AERONET sites may worsen the AOD-PM
relationship. Second, PM measurements are also more frequent (almost every day), while AERONET data are only available
with clear sky condition. Thus, the performance of CAMSRA in those days without AERONET data is unknown. Therefore,
some unknown biases or factors from CAMSRA in cloudy days may also additionally deteriorate the AOD-PM relationships.

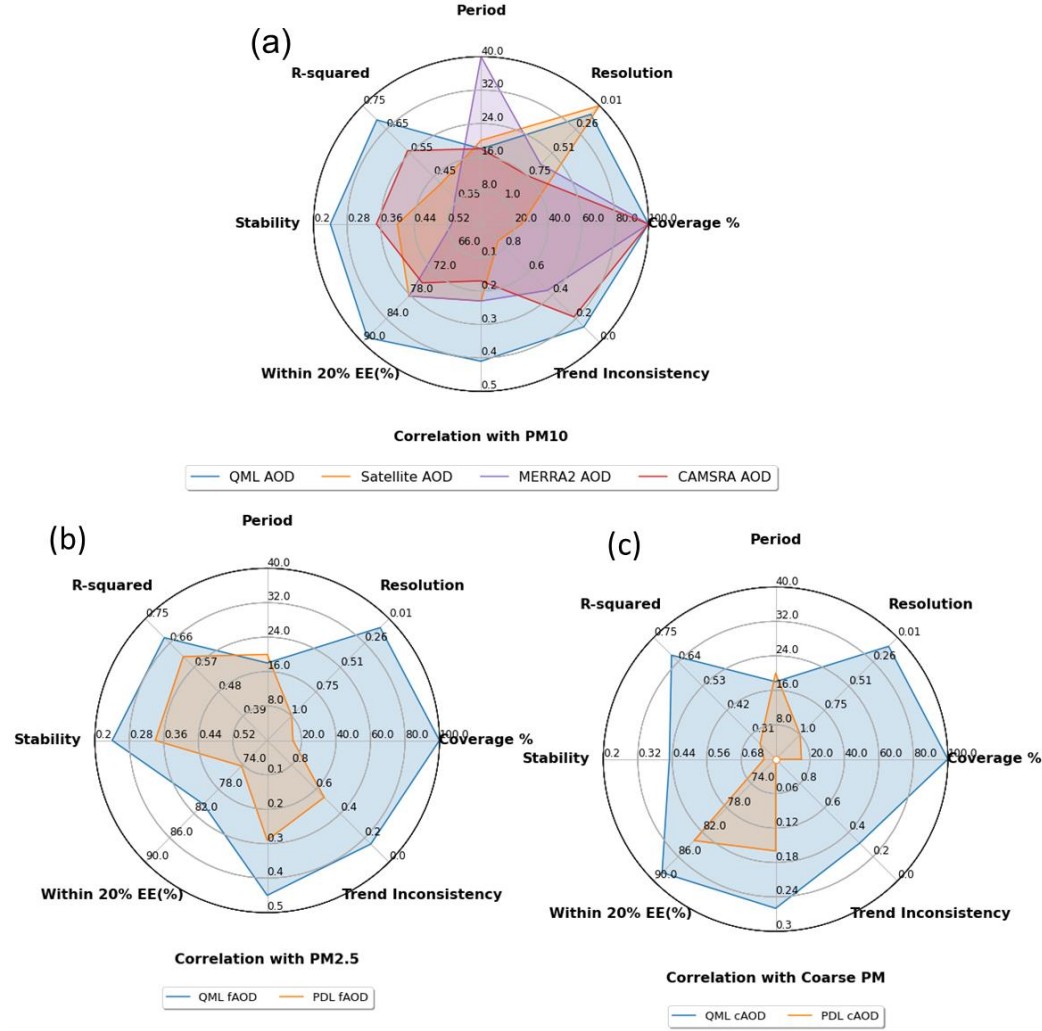

**Figure 13.** Radar plot for three different AOD product (AOD (a), fAOD (b) and cAOD (c)) with different data sources: The QML method (blue); Satellite MAIAC (fAOD and cAOD based on PDL method) (orange); MERRA-2 (green); CAMSRA (red). The eight dimensions in radar plot: Accuracy (Spatial CV $R^2$ for our methods or $R^2$ for other products); Stability (the variation coefficients of Spatial CV $R^2$ for our methods or $R^2$ for other products among different locations); the percentage (%) of bias within 20% EE; Correlation with corresponding PMs; Trend Inconsistency (TI, formula listed in Table S3); Coverage (the overall spatial coverage); Resolution (degree of resolution); Period (the data provided years).

As for the trend comparison, the satellite MAIAC AOD is least consistent with AERONet AOD, suggesting that the missing values in MAIAC AOD affects the analysis of aerosol trend. Additionally, the satellite AOD is the only product that cannot provide full temporal coverage, with only 24.21% coverage. Among these products, MERRA-2 provide the longest records (i.e., from 1980), while our data period starts from 2003, limited by the CAMSRA input.

Figure 13 (b) shows that the QML fAOD outperforming the PDL fAOD in most dimensions. The QML fAOD product is more accurate, stable and highly correlated with PM2.5 compared with PDL. QML also improves some shortcomings of PDL by



improving the spatial coverage of data from 14.96% to 100%, and increases the resolution from 1 degree to 0.1 degree. As for
the trend comparison, the QML fAOD is more consistent with AERONET data, partly because of the better coverage of the
product.

QML greatly improves the performance for cAOD (Figure 1(c)): more accurate, robust performance and more highly correlated
with Coarse PM. In the trend comparison, the QML cAOD agrees well with AERONET data, while PDL cAOD shows the
opposite trend against AERONET data (its value of TI is infinite). Thus, the QML cAOD provides a better tool for coarse aerosol
time series analysis, and its full coverage and higher resolution makes it possible to better estimate Coarse PM in the future.
**5. Conclusion**
Europe is the one of regions with the poorest association between satellite AOD and ambient PM2.5 (Christopher and Gupta,
2020).  It is a great challenge to obtain suitable aerosol products to estimate PMs in Europe. However, existing aerosol products
generally cannot provide full-coverage and reliable particles-size fraction information in high resolution. Therefore, this study
generated a new 18-years aerosol product (AOD, fAOD and cAOD) at 0.1 degree, to better understand the European different
particles-size AOD´s distribution. These three products also provide the better indicator for PM10, PM2.5 and Coarse PM,
respectively.

The out-of-sample validation of the QML AOD, fAOD and cAOD are extensively evaluated in the spatial and temporal
dimensions. Compared with other products, their NRMSE is 21%-55% lower, reaching 21.25, 21.65 and 27.56 %, respectively.
Their $R^2$ is 11-132% higher than that of other products, reaching 0.68, 0.66 and 0.65. Over 88.8, 80.5 and 88.6 % of biases
respectively fall within a ±20 % EE envelope. In correlation exploratory analysis, we found that the QML fAOD products fixed
the problem of the poor association with PM2.5, by providing higher quality and coverage predictions. The spearman correlation
almost doubles from 0.10-0.29 to 0.45. We also found that different-size PMs may be better predicted with different AOD
fractions, instead of using total AOD. For example, the QML AOD and cAOD are better indicators of PM10 and Coarse PM,
than other AOD products.

This new aerosol dataset and models not only avoid some shortcomings (e.g., lower coverage, discontinuous time) and biases,
caused by missing satellite aerosol information, but also meet the urgent need of reliable fine-mode and coarse-mode AOD data
to better estimate surface-level PMs. Thereby, it is a useful tool to monitor or to analyse the fine-mode and coarse-mode aerosols
in the spatial and temporal scales, and to further investigate their impacts on human health, the environment and the climate.
**6. Data availability**

The pan-European high-resolution aerosol optical depth (AOD) daily estimations and its fraction products developed by this
study is available at https://doi.org/10.5281/zenodo.7756570 (Chen et al., 2023). The QML AOD data are in the Geotiff format
on a daily scale.

**7.Contributions**

In this project, ZC, HP, CP, and JB conceptualized and acquired funding, while ZC and RM collected and processed the data.
ZC, HP, CP and JB contributed to the methodology. ZC and JB prepared the initial draft of the paper. ZC, HP, AL, CP and JB
contributed to the writing, review and editing. The research activity was supervised by HP, CP, and JB.





**8.Competing interests**

The contact author has declared that neither they nor their co-authors have any competing interests.

**9.Acknowledgements**

The authors gratefully acknowledge the European Centre for Medium Range Weather Forecasts, MERRA-2, ERA-5 and AERONET teams for their effort in making the data available.

**10. Financial support**

In this initial version of the geodatabase, the authors from ISGlobal would like to express their gratitude for the support they received from various organizations. The Spanish Ministry of Science and Innovation's "Centro de Excelencia Severo Ochoa 2019-2023" Program (CEX2018-000806-S-20-1), the Ministry of Research and Universities of the Government of Catalonia (2021 SGR 01563), and the Generalitat de Catalunya through the CERCA Program all provided support. CP acknowledge funding from the AXA Research Fund through the AXA Chair on Sand and Dust Storms at BSC and H2020 ACTRIS IMP (#871115). HP has received funding from the Ramon y Cajal grant (RYC2021-034511-I) and the European Union's NextGeneration EU/PRTR (PID2020-116324RA695). JB gratefully acknowledges funding from the European Union's Horizon 2020 and Horizon Europe research and innovation programs under grant agreements No 865564 (European Research Council Consolidator Grant EARLY-ADAPT) and 101069213 (European Research Council Proof-of-Concept HHS-EWS), as well as from the Spanish Ministry of Science and Innovation under grant agreement No RYC2018-025446-I (programme Ramón y Cajal).

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
