# Peer review of "A Pan-European, High-Resolution, Daily Total, Fine-Mode and Coarse-Mode Aerosol Optical Depth dataset based on Quantile 2"

_Earth System Science Data, 2023_

## Author Comment (AC1)

Review 1:

The manuscript describes the application of a supervised machine learning algorithm (lightGBM) for the retrieval of AOD, fAOD, and cAOD over Europe. However, the method presented for aerosol retrieval is not new, and I have some main concerns about this study. Firstly, the claimed high-resolution (0.1 degree) aerosol product is questionable. Secondly, the validation of the proposed model shows severe overfitting.

Thank you for taking the time to review our manuscript. We appreciate your feedback and comments. We understand your concerns regarding the novelty of the method presented for aerosol retrieval and the validation of the proposed model, however, our goal in this paper is to provide improved fAOD and cAOD products over Europe. To that end our methods focus on addressing limitations related to coarse spatial resolutions or the large data gaps of current products, rather than introducing a completely new machine learning framework. The quality of fAOD and cAOD products are key to better model particulate matter of different diameter ranges (e.g., PM2.5 and PM10), which is needed for future epidemiological studies. To make the lightGBM method more suitable for our goal, we developed three techniques, including distance weighted loss function, minimum directional distance, and white-noise data augment to improve the models. As for the other two main concerns (spatial resolution and validation), we have revised our manuscript to avoid any misunderstanding and we have done some additional sensitivity analyses to show the robustness of our predicted aerosol dataset as detailed below.

Major concerns:

1. The study claims that their AOD products were generated at a spatial resolution of 0.1 degrees. However, it should be noted that the key input variable, MAIAC AOD, only has a spatial resolution of 1km, and was eventually excluded from the models. Other variables used in the study have a lower spatial resolution than 0.1 degrees. Therefore, it is questionable whether the resulting product is truly a 0.1 degree product.

Response: Thank you for your review and for raising these concerns. We apologize for the confusion regarding the resolution of our inputs. In the submitted manuscript we inadvertently omitted specifying the resolution of 0.1 degrees of the ERA5-land surface variables, which contributed to the resolution of the final product. In the revised manuscript we included the description of the variable inputs both from ERA5 and ERA5-land (Lines 138-160), and added the information about the resolution in Table S2.

2. Additionally, Figure 11 shows that the developed AOD (B1) does not provide better details than the CAMS AOD (0.75 degrees) and MERRA-2 AOD (0.625 degrees * 0.5 degrees).

Response: Regarding the differences between QML AOD and CAMS and MERRA-2 AOD, we recognize that it is difficult to highlight the differences between products in continental maps showing 18-year averages, as originally shown in Figure 11. A zoom of Figure 11 focusing on Italy and Spain, where MAIAC AOD has fewer missing gaps, shows that the QML AOD exhibits substantially more details compared to CAMS AOD and MERRA-2 AOD (Figure R1). It is remarkable to verify that the spatial variability of the QML AOD is close to that of the MAIAC AOD, despite that MAIAC is omitted by the model. While MAIAC AOD appears to show slightly more details, likely due to its higher 1km resolution, the fewer observations available in MAIAC may also partly contribute to these differences. Figure R1A illustrates that the limited observations in MAIAC AOD can introduce some

biases when calculating 18-year averages for comparison, as opposed to solely comparing the overlapping available period of both MAIAC and AERONET data. These biases do not exist in QML AOD, because QML AOD have full coverage in 18 years.

[Figure]

Figure R1. 18-year averages (2003-2020) of different AOD in Spain(A-D) and Italy(A1-D1). We illustrate the results for these regions because MAIAC AOD has few missing gaps in southern Europe (e.g. low cloudiness). Notedly, the CAMS AOD and MERRA-2 AOD here have already been spatially interpolated to 0.1 degrees.

[Figure]

Figure R1A. The bias between MAIAC and AERONET AOD 18-year (2003-2020) averages when AERONET data is available(a); the bias between QML and AERONET AOD 18-year averages when AERONET data is available (b); the bias between MAIAC and AERONET AOD 18-year averages only when both MAIAC and AERONET data are available (c); bias between QML and AERONET 18-year averages only when both MAIAC and AERONET data are available (d). MAE is the mean absolute error.

To further validate our product, Figure 4 in the manuscript clearly demonstrates that QML AOD exhibits a better fit with AERONET AOD at the daily level compared to other products, as evidenced by the higher spatial cross-validation R square (0.68 compared to 0.36-0.52) and lower NRMSE (21.25% compared to 31.24%-32.96%).

Additionally, we have observed that most products show a good fit with AERONET AOD when calculating the 18-year averages (see Figure R2). However, the differences in performance among the four products become more pronounced when comparing AOD products with daily AERONET AOD. This suggests that while the predictions in 18-year averages may appear quite similar, their daily estimations can vary significantly. For instance, Figure R3 (randomly selected day) demonstrates that the differences among the four products become more noticeable at the daily resolution. While there is a general similarity in the patterns across these products, there are substantial differences in certain locations. Notably, QML AOD continues to exhibit more details than CAMS AOD and MERRA-2 AOD even at the daily scale.

[Figure]

Figure R2. The scatter plots of different AOD products against daily AERONET data and 18-year averaged AERONET data. Red line is y=x.

[Figure]

Figure R3. the different AOD in Europe (A-D), Spain (A1-D1) and Italy (A2-D2) in 2016-01-01.

3. Based on the input variables listed in Table S2, it appears that only the CAMS reanalysis data provides information related to aerosol size. The study seems just used the lightGBM algorithm to correct the CAMS-based fAOD and cAOD using meteorological data.

Response: Thank you for raising this question. Note that Table S2 has been updated to show the variables that come from ERA5 (0.25 degrees) and ERA5-Land (0.1 degrees). It is important to note that we do not use any explicit size information from CAMSRA. We use the separate contributions to AOD at 550 nm of "black carbon aerosol," "dust aerosol," "organic matter aerosol," "sea salt aerosol," and "sulphate aerosol, which provide size information implicitly.

We developed fAOD and cAOD products specifically to better model the particulate matter of different diameter ranges (e.g., PM2.5 and PM10), which is useful for epidemiological studies. Our sensitivity analysis indeed shows that the correlation between PM2.5 and QML fAOD is stronger than with other CAMSRA composition products (Table R1), and similar results are seen with PM10 and PMcoarse.

Table R1. The spearman correlation between different-size ground-level particulate matter (PM10, PM2.5, PMcoarse) with QML AOD products and CAMSRA composition aerosol products.

| Correlation | QML AOD | QML fAOD | QML cAOD | CAMSRA Sea Salt | CAMSRA Sulphate | CAMSRA Organic Matter | CAMSRA Dust | CAMSRA Black Carbon |
|---|---|---|---|---|---|---|---|---|
| PM2.5 | 0.40 | **0.45** | 0.02 | -0.31 | 0.16 | 0.19 | 0.11 | 0.12 |
| PM10 | **0.41** | 0.37 | 0.12 | -0.29 | 0.17 | 0.17 | 0.16 | 0.14 |
| PM coarse | 0.15 | 0.06 | **0.21** | 0.13 | 0.12 | 0.07 | 0.19 | 0.12 |

The CAMSRA compositional data contributes to the final model. However, their contribution is not more significant than other data sources. The importance score results from the lightGBM models in Figure R4 (or Figure A1 in supplementary) reveal that the contributions of each variable in the top 20 are relatively similar, ranging from around 2.8% to 4%. It is important to recognize that meteorological factors derived from ERA5 or ERA5_land, such as boundary layer dissipation (BLD) and height (BLH), humidity (RH), surface pressure (SP), and wind speed, also play a significant role in the modeling of fAOD (fine mode aerosol optical depth) and cAOD (coarse mode aerosol optical depth).

[Figure]

[Figure]

Figure R4. the top 20 important feature plots of AOD, fAOD and cAOD model, importance scores here representing the proportion of model contribution for each feature. The full name of variable is as following:

| Short name | Source | Long name |
|---|---|---|
| **MERRA_AOD** | MERRA-2 | MERRA2 aerosol optical depth 550nm |
| **TCO3** | ERA5 | total column ozone |
| **CAMS_SSAOD550** | CAMSRA | sea salt aerosol optical depth 550nm |
| **U10** | ERA5_land | 10m u component of wind |
| **V10** | ERA5_land | 10m v component of wind |
| **RH** | ERA5_land | Surface relatively humidity |
| **BLD** | ERA5 | boundary layer dissipation |
| **BLH** | ERA5 | boundary layer height |
| **SP** | ERA5_land | surface pressure |
| **CAMS_DUAOD550** | CAMSRA | dust aerosol optical depth 550nm |
| **WINDSPEED** | ERA5_land | 10m v component of wind |
| **CAMS_SUAOD550** | CAMSRA | sulphate aerosol optical depth 550nm |
| **TCC** | ERA5 | total cloud cover |
| **LCC** | ERA5 | low cloud cover |
| **CAMS_BCAOD550** | CAMSRA | black carbon aerosol optical depth 550nm |
| **D2M** | ERA5 | 2m dewpoint temperature |
| **YEAR** | Time | year |
| **DOY** | Time | day of year |
| **MSDWSWRF** | ERA5_land | Surface solar radiation downwards |
| **CAMS_OMAOD550** | CAMSRA | organic matter aerosol optical depth 550nm |

| Short name | Source | Long name |
|:---:|:---:|:---:|
| **T2M** | ERA5 | 2m temperature |
| **HCC** | ERA5 | high cloud cover |

4. It is unclear how well the developed fAOD and cAOD models perform at locations where no AERONET data is available. It is also unclear whether the study used completely independent ground-based data to test the results, such as a test site that was not used in the training process. If the Table S3 intends to show this validation, but the R2 of fAOD decreased significantly from 0.68 to 0.56 in M3, suggesting that the model may have a severe issue with overfitting.

Response: Thank you for raising this point. We acknowledge that the description of techniques to improve the models was misleading or confusing the readers. We have revised the manuscript to clarify any misunderstanding (Line 180-187). We have done two validation processes: On the one hand, we randomly selected 70% of the sites as training data for the quantile lightGBM models, additional 20% of the sites were used to optimize the model, and the rest 10% sites were completely independent test data. On the other hand, we used 5-fold cross-validation to repeat the first process, in order to test the stability of all model configurations.

The R2=0.56 mentioned in the comment corresponds to the subgroup of top 1% farthest sites, with distances to their neighbors over 463km, which were mostly located at the edge of our domain. Considering your concerns regarding the performance of the model in regions with sparsely distributed stations, we divided the top 20% of sites that were farthest from their nearest neighbors, requiring distances of at least 130.5km, and trained the model using the remaining sites. Then we compared these results to random split 20% validation datasets. We have done these sensitive comparisons for tAOD, fAOD and cAOD in Figures R4, R5 and R6, respectively. It shows that the results of our models among top 20% of farthest sites is relatively similar or a bit lower than the results in 20% random sites. It indicates models are quite stable without any severe overfitting.

[Figure]

Figure R4. the out of sample R-square of AOD model in top 20% of farthest sites (A) and 20% random validation sites (B), their corresponding scatter plots (C and D).

[Figure]

Figure R5. the out of sample R-square of fAOD model in top 20% of farthest sites (A) and 20% random validation sites (B), their corresponding scatter plots (C and D).

[Figure]

Figure R6. the performance of cAOD model in top 20% of farthest sites (left) and 20% random validation sites (right)

5. During the lightGBM-based training for fAOD and cAOD, the AERONET only provides data for fAOD and cAOD at 500nm. However, it is unclear how the model was trained to calculate fAOD and cAOD at 550nm, which is a crucial issue that the paper did not address.

Response: Thank you for bringing up this concern, we added a more detailed description (Line 104-115) on the procedure we followed to obtain the fAOD and cAOD at 550nm.
"To be comparable with the satellite and reanalysis data, the AERONET AOD data at 550 nm ($AOD_{550}$) was interpolated from the $AOD_{500}$ (Gupta et al., 2020; Duarte and Duarte, 2020). The equation (1) used for this interpolation is as follows:

$$AOD_{550} = AOD_{500} * (\frac{550}{500})^{-\propto^t} \tag{1}$$

*where* $\propto^t$ is the AERONET AOD Ångström exponent at 500nm, which is obtained from AERONET

spectral deconvolution algorithm (SDA) output. Before obtaining the $fAOD_{550}$ and $cAOD_{550}$, we first transformed the Fine mode fraction at 550 nm ($FMF_{550}$) from the 500 nm ($FMF_{500}$) using the equation (2):

$$FMF_{550} = \frac{fAOD_{500}*(\frac{550}{500})^{-\alpha f}}{AOD_{500}*(\frac{550}{500})^{-\alpha t}} = FMF_{500}*(\frac{550}{500})^{\alpha t - \alpha f} \tag{2}$$

where $\alpha^f$ is the AERONET fAOD Ångström exponent at 500nm. All of these parameters are available from AERONET SDA products. Finally, we obtained $fAOD_{550}$ and $cAOD_{550}$ by following the formula:

$$fAOD_{550} = AOD_{550}*FMF_{550} \tag{3}$$
$$cAOD_{550} = AOD_{550}*(1 - FMF_{550}) \tag{4}$$

,"

Specific concerns:

1. In Figure 1, it is not clear how to use Boruta to select the variables.

We revised the description on the use of Boruta in Line 180-187. "For each iteration, the Boruta algorithm generates a new set of shadow variables by randomly permutating the values of each potential variable, and trains a random forest classifier on the original and shadow features. The importance score of each original feature is compared to the maximum importance score of its corresponding shadow features. If the original feature has an importance score that is significantly higher than the maximum importance of its corresponding shadow features, it is considered important. Then Boruta marks the important features and removes the shadow features associated with them, and repeats these steps until a predefined number of iterations (e.g., 50 iterations in our study) have been reached."

2. The caption of the Figure 3 says "Spatial and temporal distribution of the median value of AERONet (a) AOD, (b) fAOD and (c) cAOD data". It makes me confused how (a), (b) and (c) reveal the temporal information.

Revised, thanks: "Spatial distribution of the median value of AERONet (a) AOD, (b) fAOD and (c) cAOD data".

3. AERONET in the figure caption is "AERONet", but in the text is "AERONET".

Revised.

4. Typing errors: P10, L285, (Levy et al., 2010; Xiao et al., 2016; Yan et al., 2022)).

Revised.

---

## Author Comment (AC2)

Reviewer 2:

This study presents a daily AOD data set over Europe over the period 2003-2020, which was derived by post-processing the current satellite and reanalysis products, based on Machine Learning method. The accuracy of the total AOD in this dataset has been greatly improved. At the same time, the dataset can provide additional fine/coarse AOD data, which are also relatively reliable and will be very helpful for particulate matter (PM) prediction. The dataset will be interesting for the scientific community. Therefore, I have some comments before it could be accepted for publication.

Response: Thank you for taking the time to review our study and provide your useful feedback. We appreciate your interest in our work and are happy to hear your thoughts and address any concerns you may have regarding the dataset we have presented. We believe that our dataset has the potential to be a valuable resource for the scientific community and look forward to discussing it with you further.

Major comments:

1. For the Route in the absence of satellite data, the spatial resolution of all input reanalysis of AOD data (e.g. MERRA-2, CAMS) is relatively coarse lower than 0.1 degrees, it is not appropriate to increase the spatial resolution of final AOD product to 0.1 degrees through interpolation, as simple interpolation cannot increase the AOD variation in spatial details. I think the spatial resolution of the final AOD product should not be higher than the maximum spatial resolution of one of input reanalysis data.

Response: Thank you for your review and for bringing up these concerns. We apologize for any confusion regarding the resolution of our inputs. Previously, we have not clearly described the variable inputs from ERA5 (0.25 degrees) and ERA5_land (0.1 degrees). We have made revisions to the description of the variable inputs from ERA5 and ERA5_land (Line 138-160), as well as added resolution information in Table S2. And some surface-related variables provided by ERA5-Land, like surface solar radiation downwards (MSDWSWRF), surface humidity (RH), surface wind speed (U10,V10, WINDSPEED), surface pressure (SP) are also top 20 contributed features in our final 0.1 degree resolution product (Figure R1).

[Figure]

[Figure]

Figure R1. the top 20 important feature plots of AOD, fAOD and cAOD model, importance scores here representing the proportion of model contribution for each feature. The full name of variable is as following:

| Short name | Source | Long name |
|---|---|---|
| **MERRA_AOD** | MERRA-2 | MERRA2 aerosol optical depth 550nm |
| **TCO3** | ERA5 | total column ozone |
| **CAMS_SSAOD550** | CAMSRA | sea salt aerosol optical depth 550nm |
| **U10** | ERA5_land | 10m u component of wind |
| **V10** | ERA5_land | 10m v component of wind |
| **RH** | ERA5_land | Surface relatively humidity |
| **BLD** | ERA5 | boundary layer dissipation |
| **BLH** | ERA5 | boundary layer height |
| **SP** | ERA5_land | surface pressure |
| **CAMS_DUAOD550** | CAMSRA | dust aerosol optical depth 550nm |
| **WINDSPEED** | ERA5_land | 10m v component of wind |
| **CAMS_SUAOD550** | CAMSRA | sulphate aerosol optical depth 550nm |
| **TCC** | ERA5 | total cloud cover |
| **LCC** | ERA5 | low cloud cover |
| **CAMS_BCAOD550** | CAMSRA | black carbon aerosol optical depth 550nm |
| **D2M** | ERA5 | 2m dewpoint temperature |
| **YEAR** | Time | year |
| **DOY** | Time | day of year |
| **MSDWSWRF** | ERA5_land | Surface solar radiation downwards |
| **CAMS_OMAOD550** | CAMSRA | organic matter aerosol optical depth 550nm |

| Short name | Source | Long name |
|:---:|:---:|:---:|
| **T2M** | ERA5 | 2m temperature |
| **HCC** | ERA5 | high cloud cover |

2. For the correction of total AOD, it can be understood that the information of AOD mainly comes from the AOD data of reanalysis product. But for obtaining fine AOD and coarse AOD, this study should clarify which input data plays a dominant role.

Response: Thank you for your question. We understand your concern about the dominant role of input data in obtaining fine AOD and coarse AOD. To address this, we have added a feature importance score plot as Figure S1 (the same as Figure R1 here) in the appendix. The plot lists the top 20 variables for the AOD, fAOD, and cAOD model, respectively. Interestingly, the top 10 inputs for these models are quite similar, but with a slightly different order. The top 10 inputs for all models are MERRA-2 AOD (MERRA AOD), total column of ozone from ERA5 (TCO3), CAMS sea salt AOD and dust AOD (CAMS_SSAOD550 and CAMS_DUAOD550), u and v component of wind (U10 and V10) from ERA5_land, boundary layer dissipation (BLD) and height (BLH) from ERA5, humidity (RH), and surface pressure (SP) from ERA5_land.

We found that the contributions of each variable in the top 20 were quite similar, ranging from around 2.8% to 4%. Therefore, we cannot identify a single dominant variable in the model.

3. I'm also curious, what would happen for QML AOD if two reanalysis datasets MERRA-2 and CAMS were not used as input data simultaneously?

Response: Thank you for your question. To address this concern, we conducted a sensitivity analysis in which we excluded both MERRA-2 and CAMS from our input datasets. Figure R2 showed a decrease in model performance for AOD, fAOD, and cAOD in the test sites, with the correlation coefficients decreasing from 0.71 to 0.47, from 0.69 to 0.45, and from 0.70 to 0.43, respectively. It suggested that air quality reanalysis data indeed contributes around one third to the information of the model, while other sources of data, such as meteorological data from ERA5 or surface data from ERA5 land, also contribute to some degree to the model. Figure R2 (a1-c1) also suggested that excluding both MERRA-2 and CAMS as input datasets could lead to the underestimation of the higher AOD values.

[Figure]

Figure R2. Comparison of the original models (a-c) and the model without MERRA-2 and CAMS data (a1-c1): AOD (a), fAOD (b) and cAOD (c).

Minor comments:

1. In section 2, this manuscript should introduce the basic information of PM data, as it was used in subsequent experiments.

Response: Done, We added in Line 161-167.

2. Line 105, how about fAOD and cAOD at 550nm was interpolated?

Response: Thank you for bringing up this concern, we added a more detailed description (Line 104-115) on the procedure we followed to obtain the fAOD and cAOD at 550nm: "To be comparable with the satellite and reanalysis data, the AERONET AOD data at 550 nm ($AOD_{550}$) was interpolated from the $AOD_{500}$ (Gupta et al., 2020; Duarte and Duarte, 2020). The equation (1) used for this interpolation is as follows:

$$AOD_{550} = AOD_{500} * \left(\frac{550}{500}\right)^{-\propto^t} \tag{1}$$

where $\propto^t$ is the AERONET AOD Ångström exponent at 500nm, which is obtained from AERONET spectral deconvolution algorithm (SDA) output. Before obtaining the $fAOD_{550}$ and $cAOD_{550}$, we first transformed the Fine mode fraction at 550 nm ($FMF_{550}$) from the 500 nm ($FMF_{500}$) using the equation (2):

$$FMF_{550} = \frac{fAOD_{500}*\left(\frac{550}{500}\right)^{-\propto^f}}{AOD_{500}*\left(\frac{550}{500}\right)^{-\propto^t}} = FMF_{500} * \left(\frac{550}{500}\right)^{\propto^t - \propto^f} \tag{2}$$

where $\propto^f$ is the AERONET fAOD Ångström exponent at 500nm. All of these parameters are available from AERONET SDA products. Finally, we obtained $fAOD_{550}$ and $cAOD_{550}$ by following the formula:

$$fAOD_{550} = AOD_{550} * FMF_{550} \tag{3}$$
$$cAOD_{550} = AOD_{550} * (1 - FMF_{550}) \tag{4}$$

"

Line 108, I believe the MODIS MAIAC data that the manuscript used is Collection 6 (C6), not v6.1, as the C6.1 product (MCD19A2) has not yet completed production.

Response: Thanks for correction. we revised the sentence according to the comment of the referee

3. Line 155, how is the MODIS 1km AOD product made to 0.1 degrees?

Response: We used the area-weighted averages within 1km grids. The 1km pixels within the 10km grid cells are averaged using a weighted average approach based on the fraction of the 1km pixel that falls within the 10km grid cell.

4. Line 269, the description is not clear about"Sat scenario"and "Non-Sat scenario", what do these two words mean? How to distinguish"Sat scenario"and "Non-Sat scenario"?

Response: We revised the text to clarify the issue (Line 294-300): "To account for the large fraction of missing values in satellite MAIAC AOD data (64%), we divided the

validation results into two subgroups based on the availability of satellite MAIAC AOD data. The first subgroup, referred to as the "Sat scenario", included validation dates and sites where satellite MAIAC AOD data were available. The second subgroup, referred to as the "Non-Sat scenario", included validation dates and sites where satellite MAIAC AOD data were not available. This division was made in order to ensure comparability among the different products and to provide insights into the factors that limit the performance of the models."

5. Line 391, how was EE=±0.025 ±20 %/40 % determined? I think most literature uses 0.05 instead of 0.025.

Response: Thank you for bringing this to my attention. For total AOD, we also applied the standard EE=±0.05 ±20 %/40 % as most previous studies. The ±0.05 here represents expected error which includes measurement errors and other uncertainties that may come from instrument calibration or atmospheric conditions. Since the 0.05 value is around the 10th percentile of total AOD, and around ±5% for those larger AOD values (AOD >1), this EE can work well in most percentiles. However, fAOD and cAOD represent a subset of total AOD, and are thus generally less variable than total AOD. The 0.05 value is around the 40th percentile of fAOD, and it seems to be too big for small percentiles, while 0.025 is around the 10th percentile of fAOD. Thus, we used a narrower EE, like ±0.025 ±20 %/40 %, to ensure that the validation results are reliable in small percentiles, while still providing adequate evaluation in larger AOD values. In summary, the determination of the specific values for the EE used in this study was based on a consideration of the variability and distribution of AOD/fAOD/cAOD values in our dataset, and the will to balance the need for a reliable evaluation in small percentiles while still providing adequate evaluation in larger AOD values.

---

## Author Comment (AC3)

The dataset of AOD, fAOD and cAOD over Europe has application value for environment analysis. The machine learning method was used to produce daily AODs. The manuscript should be revised before considering publication.

Response: Thank you for taking the time to review our study and provide your useful feedback. We appreciate your interest in our work and are happy to hear your thoughts and address any concerns you may have regarding the dataset we have presented. We believe that our dataset has the potential to be a valuable resource for the scientific community and look forward to discussing it with you further. Thank you again for your thoughtful review.

General comments:

1 The spatial and temporal resolution of all input and output data for the machine learning should be listed.

Response: We appreciate your valuable feedback regarding the description of the spatial and temporal resolution of the data used for machine learning. We have made the necessary revisions to the description of the variable inputs from ERA5 and ERA5_land (Line 138-160):

**"2.5 ERA5 reanalysis for atmospheric meteorological data**

Previous studies (Huang et al., 2007; Zhou and Savijärvi, 2014; Tai et al., 2010; Gui et al., 2019; Yan et al., 2022) have analysed the associations between weather conditions and the concentration of fine- and coarse-mode aerosols. For example, high-pressure events, characterised by atmospheric stability and low winds, retain the smaller particles, which is seen with higher-than-normal fine-mode aerosol levels (Tai et al., 2010; Gui et al., 2019). Moreover, rainfall washes out the particles from the lower part of the troposphere, especially the largest particles. There are other pathways by which aerosols can also affect weather conditions, for example by reflecting and absorbing the incoming UV radiation (Zhou and Savijärvi, 2014), or by changing the conditions for the condensation of water in the cloud(Huang et al., 2007). To account for the impact of meteorological factors on aerosols, we collected data from several atmospheric variables, such as boundary layer height, downward UV radiation, cloud cover, and precipitation, from the ECMWF ERA-5 reanalysis. This dataset provides data since 1950 at a resolution of 0.25º x 0.25º. More information about the resolution and data source for each meteorological variable can be found in Table S2.

**2.6 ERA5 land reanalysis for surface data**

Apart from atmospheric meteorological data, the surface data also has important impacts on aerosol. As forests contribute to a large extent to particle removal, previous studies found the deposition velocity of ultrafine particles is generally more sensitive to leaf area index than leaf area density (Lin et al., 2018; Huang et al., 2015). Also, the dry deposition of particles is affected by properties of the vegetation elements (such as leaves and branches) and soil types (Grönholm et al., 2009). Thus, we found the significant contributions of leaf

area index high vegetation, leaf area index low vegetation and soil types to aerosol. Higher Leaf area index high vegetation means more evergreen trees, deciduous trees or forest, while Higher Leaf area index low vegetation represents more crops and mixed farming, grass or shrubs. For bare ground or places with no leaves, both of them will be close to zero. The soil types describe how coarse the soil is, representing the water holding ability of soil. Coarser soil generally has lower water holding ability. Additionally, land surface information is essential for surface reflectance, which further affects the quality of satellite data included in reanalysis data. Most surface-related variables, including some near-ground meteorological data, are provided by ERA5-Land at a resolution of 0.1° x 0.1°."

We have also added resolution information in Table S2.

**Table S2**. The list of variables used in this study

| SHORT NAME | SOURCE | RESOLUTION | LONG NAME | UNIT |
|---|---|---|---|---|
| U10 | ERA5_land | 0.1° | 10m u component of wind | m s**-1 |
| V10 | ERA5_land | 0.1° | 10m v component of wind | m s**-1 |
| RH | ERA5_land | 0.1° | Surface relatively humidity | (0 - 100) |
| LAI_HV | ERA5_land | 0.1° | leaf area index high vegetation | m**2 m**-2 |
| LAI_LV | ERA5_land | 0.1° | leaf area index low vegetation | m**2 m**-2 |
| MSDWSWRF | ERA5_land | 0.1° | Surface solar radiation downwards | J m**-2 |
| ASN | ERA5_land | 0.1° | snow albedo | (0 - 1) |
| SP | ERA5_land | 0.1° | surface pressure | Pa |
| TE | ERA5_land | 0.1° | total evaporation | m |
| D2M | ERA5 | 0.25° | 2m dewpoint temperature | K |
| T2M | ERA5 | 0.25° | 2m temperature | K |
| BLD | ERA5 | 0.25° | boundary layer dissipation | J m**-2 |
| BLH | ERA5 | 0.25° | boundary layer height | m |
| HCC | ERA5 | 0.25° | high cloud cover | (0 - 1) |
| TCC | ERA5 | 0.25° | total cloud cover | (0 - 1) |
| LCC | ERA5 | 0.25° | low cloud cover | (0 - 1) |
| SLT | ERA5 | 0.25° | soil type | 1-7, higher is finer soil with stronger ability contains water |
| MCC | ERA5 | 0.25° | medium cloud cover | (0 - 1) |
| TCO3 | ERA5 | 0.25° | total column ozone | J m**-2 |
| TP | ERA5 | 0.25° | total precipitation | m |
| ALUVD | ERA5 | 0.25° | uv visible albedo for diffuse radiation | (0 - 1) |
| ALUVP | ERA5 | 0.25° | uv visible albedo for direct radiation | (0 - 1) |
| YEAR | Time | \ | Year | \ |
| DOW | Time | \ | day of week | \ |
| DOY | Time | \ | day of year | \ |
| LAT | Spatial | \ | latitude | \ |
| LON | Spatial | \ | longtitude | \ |
| NE | Minimum directional distance | 0.1° | minimum distance to nearest sites in North-east direction | m |
| SE | Minimum | 0.1° | minimum distance to nearest | m |

| SHORT NAME | SOURCE | RESOLUTION | LONG NAME | UNIT |
|---|---|---|---|---|
| | directional distance | | sites in South-east direction | |
| SW | Minimum directional distance | 0.1° | minimum distance to nearest sites in South-west direction | m |
| NW | Minimum directional distance | 0.1° | minimum distance to nearest sites in North-west direction | m |
| CAMS_BC AOD550 | CAMSRA | 0.75° x 0.75° | black carbon aerosol optical depth 550nm | \ |
| CAMS_DU AOD550 | CAMSRA | 0.75° x 0.75° | dust aerosol optical depth 550nm | \ |
| CAMS_O MAOD550 | CAMSRA | 0.75° x 0.75° | organic matter aerosol optical depth 550nm | \ |
| CAMS_SS AOD550 | CAMSRA | 0.75° x 0.75° | sea salt aerosol optical depth 550nm | \ |
| CAMS_SU AOD550 | CAMSRA | 0.75° x 0.75° | sulphate aerosol optical depth 550nm | \ |
| MERRA_A OD | MERRA-2 | 0.625°×0.5° | MERRA2 aerosol optical depth 550nm | \ |

Due to the different resolutions of each data, the method of spatio-temporal matching should be clarified.

To address your concern about the method of spatio-temporal matching, we have provided additional details and clarification (Line 175-180) in the revised version:

"In order to address the different spatial resolutions, we employed bilinear resampling to standardize all gridded data to a horizontal resolution of 0.1° x 0.1° (equivalent to approximately 9 km at mid-latitudes). Subsequently, we extracted the corresponding values at the grid cell where the AERONET sites located. Regarding the temporal resolution, we computed daily averages for each product. This involved utilizing all available data points for a specific day to calculate the average. For example, we used hourly data from MERRA-2, ERA5, and ERA5-land, while CAMSRA data was available at a 3-hourly resolution. The AERONET data, however, was obtained at a daily frequency."

Thank you for bringing this to our attention, and we hope that the revised description provides a clearer understanding of the spatial and temporal matching methodology employed in our study.

2 As the satellite AOD was given up, I think all the inputs are reanalysis data. So the temporal resolution of AOD, fAOD and cAOD is not necessary daily. Then, which one or some certain times in one day were selected to produce daily AOD, fAOD and cAOD? And Why?

Response: Thank you for your feedback and the question regarding the temporal resolution of AOD, fAOD, and cAOD in our study. In our study, we purposely chose to use daily averages for the AOD, fAOD, and cAOD products.

The decision to use daily averages was made based on the intended future application of this AOD product, which is to estimate ground-level PM2.5 and PM10 on a daily level.

Additionally, short-term health impact assessment studies typically focus on air pollutant exposure at daily scales, as it aligns with the daily scale of health data used in such studies.

To obtain the daily averages for each product, we generally took all available data for a given day and calculated the average. For example, we used hourly data for MERRA-2, ERA5, and ERA5-land, 3-hourly data for CAMSRA, and daily data for AERONET to obtain the daily averages for each product, we generally took all available dataset on that day to calculate it, for example, using hourly data for MERRA-2, ERA5 and ERA5 -land, 3-hourly data for CAMSRA and daily data for AERONET.

We believe that selecting daily averages provides a suitable temporal resolution for our study's objectives and future applications related to air pollution exposure and health impact assessment. We appreciate your feedback and the opportunity to clarify our approach.

3 Why chose LightGBM from kinds of machine learning methods? Decision-tree based machine learning methods would adopt some fixed thresholds, which may create systematic "boundary" in the product. For example, if the latitude was included in the input data, you can see a AOD systematic boundary at a latitude line. Other parameters has the similar affects.

Response: Thank you for your feedback regarding the choice of LightGBM as the machine learning method in our study. We appreciate your concern about decision-tree based methods potentially creating systematic boundaries due to fixed thresholds.

The decision to use LightGBM was based on several factors that make it suitable for our specific application. Firstly, LightGBM is known for its high computational efficiency, making it well-suited for handling large-scale datasets like the 18-year predictions for the whole of Europe. Its faster training and prediction times on large datasets outperform other gradient frameworks, random forests, and support vector machines (SVM).

Secondly, LightGBM incorporates a gradient-based One-Side Sampling (GOSS) technique, which helps the model prioritize important data points to capture the general pattern. This prioritization, along with early stopping and regularization techniques, helps prevent overfitting and ensures good generalization performance.

Furthermore, the LightGBM framework offers a wide range of customizable APIs and parameters. This allows us to customize our own loss function and to obtain quantile predictions, which are useful for improving the model structure and estimating uncertainty in our predictions.

Regarding the concern about systematic boundaries, it is true that a single decision tree can create clear boundaries in predictions based on fixed thresholds. However, both Random Forest and LightGBM ensemble multiple decision trees, which helps to blur these

boundaries. By averaging the predictions of multiple trees or using gradient boosting to correct errors, the ensemble methods can reduce the impact of fixed thresholds and produce more flexible decision boundaries. Additionally, considering more variables and interactions can enable the model to capture more complex patterns and reduce the influence of fixed thresholds. As shown in Figure 11 of our study, we observed that no systematic "boundary" is prominent in our product. In the future, we plan to explore kernel-based machine learning methods as well to further mitigate the potential systematic boundary problem.

4 The spatial distribution, I am not sure if it means some AERONET sites data were not used in training, and only used in test? If so, that's real spatial independent validation. If not, we can not give the accuracy over locations which has no AERONET site.

Response: Thank you for your feedback and the question regarding the spatial distribution and validation of our model. We appreciate your concerns and the opportunity to clarify the validation process. We have revised the manuscript (Line 196-198) and (Line 220-227) to provide a more detailed explanation.

In our study, we conducted two validation processes to assess the performance of the model. Firstly, we randomly selected 70% of the AERONET sites as training data for the quantile LightGBM models. An additional 20% of the sites were used to optimize the model, and the remaining 10% of the sites served as completely independent test data. Table S1 in our study presents the results, showing that the R-squared values for the independent test sites are 0.72, 0.69, and 0.70 for AOD, fAOD, and cAOD, respectively.

The second validation process involved using 5-fold cross-validation, which repeated the first process multiple times to test the stability and consistency of the model configurations. Table S1 also presents the results for the cross-validating test sites, indicating R-squared values ranging from 0.68 to 0.74 for AOD, 0.65 to 0.73 for fAOD, and 0.68 to 0.74 for cAOD. These values are similar to the results obtained in the first process.

Furthermore, Table S3 in our study compares the results between randomly selecting test sites and using the top 20% of sites that are farthest from their nearest neighbors as test sites. The small differences observed between these two situations further indicate the robustness of our model, even in locations far away from AERONET sites.

In summary, our validation processes and results demonstrate that the framework of our model is robust and capable of providing accurate predictions even in locations without AERONET sites. We appreciate your feedback and the opportunity to clarify the validation procedures.

Minor comments:

1 The abbreviation should be explained at the first appearance, such as "NMB" in the supplement.
Added.

2 The section numbers are wrong in chapter 4.

Revised.